# Structural basis of supercoiling-induced CRISPR–Cas9 off-target activity

Quentin M. Smith[1,2], Sylvia Whittle[3], Ricardo J. Aramayo[1], Daniel E. Rollins[3], Adam S. B. Jalal[2], Deborah I. Egharevba[1], Kyle L. Morris[4], Alice L. B. Pyne[3✉] & David S. Rueda[1,2✉]

CRISPR–Cas9 is a powerful genome-editing tool[1], but genome-wide off-target activity can hinder therapeutic applications. Negative supercoiling ((−)SC) has been implicated in off-target activity, but a molecular-level understanding is lacking. Here, using (−)SC DNA minicircles, we observe supercoiling-driven structural defects in the DNA that are resolved by Cas9 binding. Cryo-electron microscopy structures of Cas9 bound in both the on-target and off-target configurations highlight that the Cas9 HNH domain is poised in a more catalytically competent conformation. New DNA–RNA mismatch geometries are accommodated across the protospacer and structural plasticity in the protospacer adjacent motif distal region of the protospacer is topology dependent. Together, our study reveals the molecular basis for (−)SC-induced Cas9 targeting and provides a framework for the design of next-generation high-fidelity CRISPR effectors with topological context.

Clustered regularly interspaced short palindromic repeats (CRISPR) and CRISPR-associated proteins (Cas) arose as a bacterial defence mechanism against viral infections. CRISPR–Cas systems have been harnessed as powerful genomic editing tools for eukaryotic cells[1]. *Streptococcus pyogenes* Cas9 (*Sp*Cas9) is an RNA-guided endonuclease that recognizes a three-nucleotide protospacer adjacent motif (PAM, 5′-NGG-3′), then unwinds the 20-base-pair (bp) protospacer target to form a DNA–RNA hybrid[2,3]. Previous studies have shown that complementarity between the guide RNA (gRNA) and the protospacer results in double-stranded DNA cleavage by two endonucleolytic domains, HNH and RuvC. However, alternative PAM recognition and imperfect complementarity can lead to undesired off-target editing. Off-target edits represent an important barrier to therapeutic applications. Various approaches identify off-target effects with increasing sensitivity, demonstrating the genome-wide off-targeting landscape of *Sp*Cas9[4]. Numerous biochemical[5,6], structural[7,8] and single-molecule experiments[9–12] have provided insights into the molecular mechanism of off-target activity.

DNA topology has been shown to strongly modulate Cas9 off-target activity. Biochemically or mechanically unwinding DNA enhances the off-target activity of thermophilic Cas9 effectors[3,13–15], on isolated targets and genome-wide[16]. In cells, DNA unwinding is regulated by topoisomerases[17] and has been linked to several physiological processes, such as gene expression[18,19], chromatin structure[20,21] and DNA replication[22,23] and repair[24,25]. To decrease off-target activity, high-fidelity Cas9 variants have been engineered through directed evolution screens[26,27] or using structural insights[7,11,28]. However, how DNA topology modulates Cas9 targeting at the molecular level remains unclear due to a lack of structural information on (−)SC substrates. It is therefore essential to understand the interplay between DNA topology and Cas9 activity to devise improved Cas9 variants.

To elucidate the role of (−)SC in Cas9 target recognition, allosteric activation and cleavage, we have solved a set of cryo-electron microscopy (cryo-EM) structures of dead Cas9 (dCas9) bound to (−)SC DNA minicircles. These structures show that, bound to on-target (−)SC DNA minicircles, dCas9 resembles linear DNA structures but poises the HNH nuclease domain 15 Å closer to the target strand (TS) scissile phosphate. Our (−)SC off-target structures show mismatch accommodation across the protospacer, including several in the seed region, with to date undescribed R-loop base-pair geometries. R-loop flexibility in the PAM distal region substantially increases the tolerance for mismatches and reduces the R-loop requirement for cleavage to just 14 bp. Finally, we use a combination of optical tweezers and fluorescence resonance energy transfer (FRET) to capture the HNH domain dynamics when bound to on- and off-target (−)SC λ-DNA. Together, our data reveal the molecular basis of Cas9 targeting in the presence of (−)SC, laying the foundation for the design of the next generation of high-fidelity CRISPR effectors.

## Cas9 resolves (−)SC-induced defects

DNA minicircles are versatile substrates for DNA topology studies[29–32]. Using splinted ligation (Extended Data Fig. 1a), we have generated 126 bp DNA minicircles containing the well-characterized EMX1-1 target sequence (Fig. 1a,b) and two off-target sequences selected from our previous data based on mismatch number, distribution and identity[16] (Extended Data Fig. 1b,c). These minicircles were negatively supercoiled using DNA gyrase to a linking difference, $\Delta Lk = -2$, corresponding to a supercoiling density ($\sigma$) = −0.167. (−)SC DNA minicircles exhibit a band shift relative to relaxed or nicked minicircles, as expected (Fig. 1a). Restriction-enzyme-treated minicircles yield a faster migrating band, confirming their size (Fig. 1a).

[1]MRC Laboratory of Medical Sciences, Imperial College London, London, UK. [2]Department of Infectious Disease, Imperial College London, London, UK. [3]Department of Materials Science and Engineering, University of Sheffield, Sheffield, UK. [4]Wellcome Genome Campus, European Molecular Biology Laboratory, European Bioinformatics Institute (EMBL-EBI), Hinxton, UK. ✉e-mail: a.l.pyne@sheffield.ac.uk; david.rueda@imperial.ac.uk

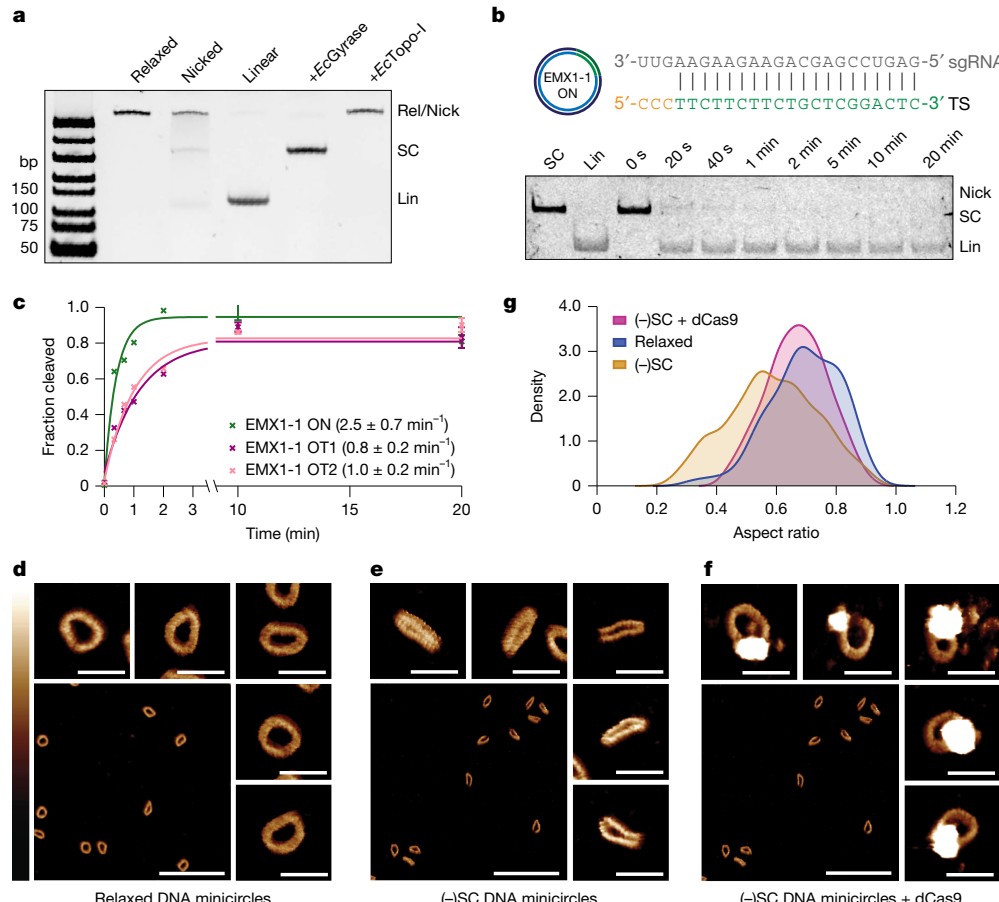

**Fig. 1 | Synthesis and characterization of (−)SC DNA minicircles. a**, 10% native PAGE analysis, showing relaxed (Rel), nicked (Nick), linearized (Lin) and (−)SC DNA (SC) minicircles, as well as *Escherichia coli* topoisomerase-I (*Ec*Topo-I)-treated (−)SC DNA as a control. **b**, On-target (−)SC DNA minicircles are rapidly cleaved by Cas9, generating linearized product within 20 s. **c**, The cleavage time course yields and apparent cleavage rate ($k_{cleave}$ (on target (ON)) = 2.5 ± 0.7 min$^{-1}$). Cas9 also cleaves off-target (OT) (−)SC DNA minicircles with apparent rate constants of $k_{cleave}$ (OT1) = 0.8 ± 0.2 min$^{-1}$ and $k_{cleave}$ (OT2) = 1.0 ± 0.2 min$^{-1}$. Data are mean ± s.e.m. **d**, High-resolution AFM images of relaxed ON DNA minicircles. **e**, High-resolution AFM images of ON (−)SC DNA minicircles. **f**, High-resolution AFM images of ON (−)SC DNA minicircles complexed with dCas9. For **d**–**f**, scale bars, 100 nm (wide view) and 20 nm (individual molecules). Height scale, 0–3 nm. **g**, Kernel density estimate plot of aspect ratios of species described in **d**–**f** (*n* = 1,000, 3 experimental repeats).

To confirm that these minicircles are bona fide Cas9 substrates, we performed cleavage assays using Cas9 and (−)SC DNA minicircles encoding on- and off-target sequences. Cas9 rapidly cleaves the on-target (−)SC DNA minicircles (Fig. 1b) to a linear form in <20 s ($k_{obs} ≥ 2.5 ± 0.6$ min$^{-1}$), tenfold faster than with a linear substrate[8], but does not cleave linear off-target substrates (Extended Data Fig. 1h–k). Cas9 also cleaved the off-target minicircles (OT1 and OT2; Fig. 1c and Extended Data Fig. 1b,c), albeit around threefold slower than the on-target minicircles. Despite numerous seed-region mismatches in OT1, Cas9 cleaves both off-target sequences at similar rates. This is probably because (−)SC lowers the local melting energy barriers, thereby facilitating R-loop formation and increasing mismatch tolerance, even in the seed region. Notably, while the overall off-target cleavage rates are near-identical (Fig. 1c), OT1 exhibits a nicked intermediate but OT2 does not (Extended Data Fig. 1b), indicating that the first cleavage event is faster for OT1 but not for OT2.

We selected two additional off-target sequences (OT3 and OT4) with distinct mismatch combinations, which are also cleaved in (−)SC but not in the linearized form (Extended Data Fig. 1d–k). These results are consistent with the idea that cleavage enhancement by (−)SC substrate is a general, sequence-independent effect. However, the target sequence can still modulate the cleavage rates. Notably, the off-target cleavage rate constants with (−)SC minicircles are generally 100-fold faster than on linear off-target sequences[8].

To define the role of DNA circularization on R-loop formation, we performed electrophoretic mobility shift assays (EMSAs) with dCas9 on relaxed and (−)SC DNA minicircles (Extended Data Fig. 1l,m). Stable dCas9 binding is observed with (−)SC DNA, but not relaxed minicircles, confirming that (−)SC, not circularization, enhances Cas9 binding and off-target cleavage.

To understand the conformational changes induced by (−)SC, we analysed the structure of relaxed and supercoiled minicircles using high-resolution atomic-force microscopy (AFM)[31,33] (Fig. 1d–f and Extended Data Figs. 2 and 3). The relaxed and (−)SC species have contour lengths of 39.0 ± 3.2 and 34.6 ± 4.0 nm, respectively, within 4.0 and 8.4 nm of their expected lengths (Extended Data Fig. 3). Relaxed DNA minicircles appear mostly open circular (Fig. 1d), with a minimum width of 10.7 ± 1.5 nm and an aspect ratio of 0.7 ± 0.1 nm (Fig. 1g). By contrast, (−)SC DNA minicircles appear primarily as elongated structures (Fig. 1e) with a reduced minimum width of 8.2 ± 1.8 nm and aspect ratio of 0.6 ± 0.2 nm (Fig. 1g and Extended Data Fig. 3g). Most (−)SC DNAs appear to contain at least one defect site, observable as a point of lower intensity or higher bending (Fig. 1e). These observations agree with previous molecular dynamics (MD) simulations describing (−)SC DNA minicircles as double denatured[34]. Moreover, this structural collapse is consistent with the expected absence of writhe in small minicircles of a single persistence length[17].

To determine how (−)SC promotes Cas9 binding, (−)SC minicircles were imaged using AFM in the presence of dCas9. dCas9 binding was

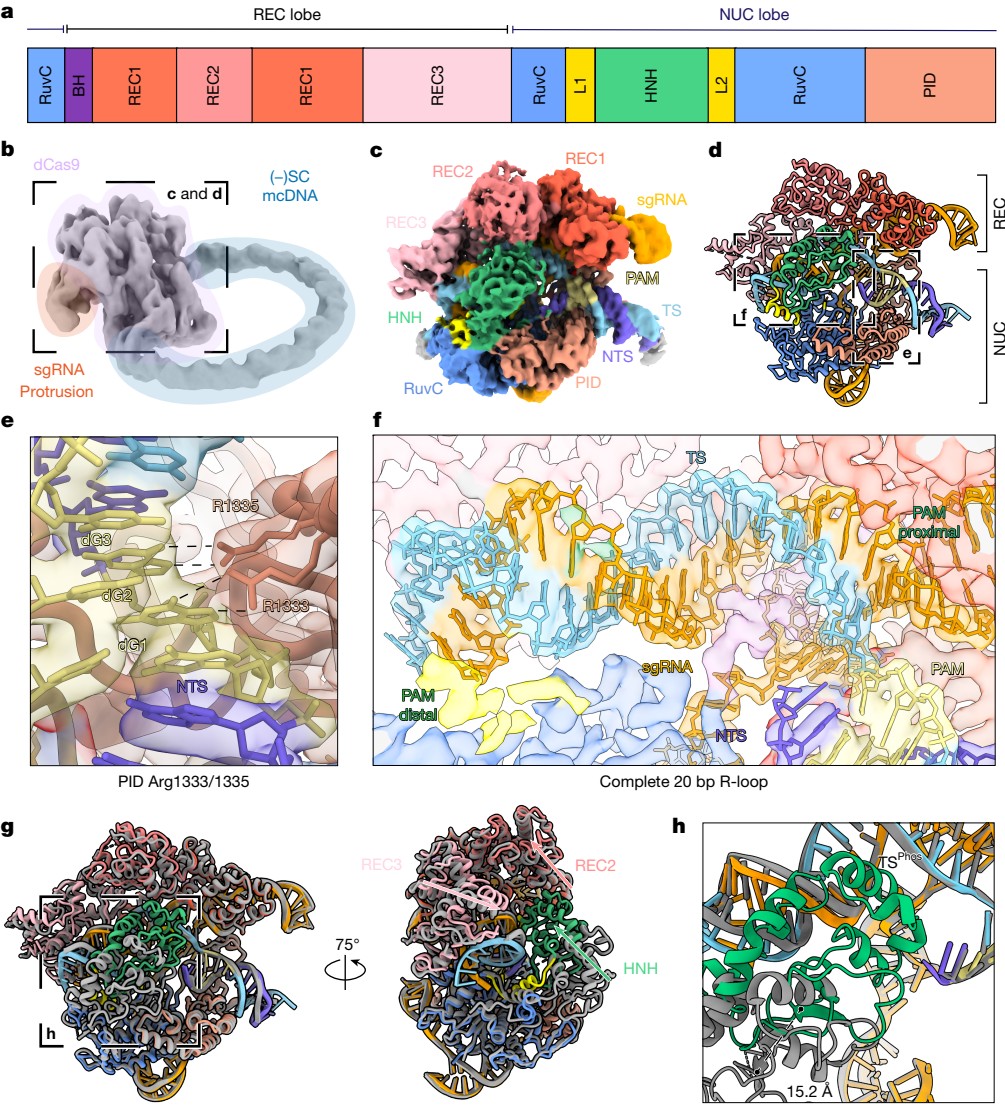

**Fig. 2 | Cryo-EM map and molecular model of dCas9 complex with on-target (−)SC DNA minicircle. a**, The domain organization of *Sp*Cas9, highlighting the recognition (REC1–3) and nuclease (HNH and RuvC) lobes, the PID, the bridge-helix (BH) and linker domains (L1, L2). **b**, Cryo-EM map with highlighted features of the complete (−)SC DNA minicircle in a complex with dCas9 (4.4 Å resolution). mcDNA, minicircle DNA. **c**, Focus-refined cryo-EM map of the dCas9 core complex (3.1 Å resolution). **d**, Molecular model of dCas9 core complex. **e**, 5′-NGG-3′ PAM recognition by PID residues Arg1333 and Arg1335.

**f**, Cryo-EM map and molecular model of the complete 20 bp R-loop. **g**, A comparison of the (−)SC DNA minicircle–dCas9 structure (this study, coloured) with the previous precatalytic structure (PDB: 6O0Z, grey) shows the allosteric movement of the HNH, REC2 and REC3 domains. **h**, Distance mapping of the catalytic residue H840A to the TS scissile phosphate for both structures; the structure in this study shows a 15 Å swing of the HNH domain to the TS scissile phosphate.

clearly visible as a 710 ± 340 nm³ globular structure on the DNA minicircle (Fig. 1f and Extended Data Fig. 3q). dCas9-bound (−)SC minicircles adopted an open circular (relaxed) structure with an aspect ratio of 0.7 ± 0.1 nm (Fig. 1g), comparable to the relaxed minicircles. This global conformational change is consistent with the change of twist observed after dCas9 binding in magnetic tweezer experiments[3,15] and is as expected for the formation of an R-loop structure within a DNA minicircle with 12 helical turns and an Δ*Lk* of −2.

To determine the structural basis for these changes, we require a higher resolution of the protein structure and, therefore, used cryo-EM.

## Cas9 structure on on-target (−)SC minicircles

Numerous Cas9 structural studies using linear double-stranded DNA (dsDNA) substrates have captured various conformational and catalytic

states along the reaction pathway[35–39]. To understand the molecular basis for how DNA topology affects dCas9 binding and allosteric activation, we determined the cryo-EM structure of dCas9 bound to a (−)SC minicircle encoding the on-target sequence (Fig. 2 and Extended Data Fig. 4). First, we obtained a 4.4 Å density map of dCas9 bound to the entire (−)SC minicircle (Extended Data Table 1). Similar to reported structures, dCas9 appears well-ordered, with the characteristic bilobed configuration and with the single guide RNA (sgRNA) protruding from the recognition (REC) lobe (Fig. 2b). Notably, the map exhibits clear density for the entire DNA minicircle ring, reminiscent of a diamond ring, with recognizable major and minor grooves (Supplementary Video 1). Consistent with the AFM results and the previous magnetic-tweezer experiments[3,15], the ring adopts an open circular (relaxed) conformation without apparent structural defects (Fig. 2b). Farthest from the dCas9 density, the DNA ring density appears weaker, reflected in the local

resolution of this part of the structure (Extended Data Fig. 4d). To better resolve the core dCas9 complex, we performed a focused refinement, yielding a second reconstruction at a resolution of 3.1 Å, permitting atomistic modelling of the core complex (Fig. 2c,d). The resulting structure exhibits all of the expected features of the core complex, including the REC and nuclease (NUC) lobes, sgRNA, TS, non-target strand (NTS), DNA–RNA hybrid (R-loop) and the PAM-interacting domain (PID).

PAM recognition is the first step in the reaction pathway of Cas9[40]. In the (−)SC minicircle structure, we observed clear PAM recognition between the PID residues Arg1333 and Arg1335 hydrogen bonding to dG2 and dG3 in the PAM (Fig. 2e), in agreement with earlier dCas9 structures[36]. The next steps in the reaction pathway involve DNA unwinding and R-loop formation[40]. Our structure reveals a complete R-loop (Fig. 2f), with each of the 20 bp resolved.

The (−)SC minicircle dCas9 structure also reveals several salient differences in comparison to previous structures. Notably, the (−)SC DNA minicircle dCas9 structure reveals an approximately 15 Å swing of the HNH domain closer to TS scissile phosphate (Fig. 2g,h and Supplementary Video 2) compared with a previous precatalytic Cas9 state structure (Protein Data Bank (PDB): 6O0Z)[38]. In parallel, the REC2 domain moves away from the TS, while the REC3 domain clamps onto the PAM distal end of the R-loop (Fig. 2g and Supplementary Video 2). Consistent with these observations, MD simulations have postulated that the REC domains and the HNH nuclease domain must undergo a concerted structural rearrangement that has an important role in allosteric activation for DNA cleavage[41]. A similar conformational swing of the HNH domain was also observed in an earlier crystal structure of dCas9 bound to a single-stranded DNA (ssDNA) target (root mean squared displacement (r.m.s.d.) of 1.14 Å for 127 Cα versus PDB 4OO8)[35]. Akin to our structure, a ssDNA target can be considered to be highly (−)SC DNA, further supporting the idea that a more labile DNA target facilitates R-loop formation, bringing the HNH active site closer to the scissile phosphate, thereby activating Cas9 for catalysis. In contrast to the HNH domain, the RuvC domain appears in a near identical conformation to in previous structures, with a r.m.s.d. of 4.5 Å for 257 Cα versus PDB 6O0Z.

Moreover, we observed substantial structural differences in the PAM distal region of the R-loop (Extended Data Fig. 5e). While the seed region is near-identical to that of the previous precatalytic Cas9 state structure[38] (0.8 Å r.m.s.d. across the first 10 bp), the PAM distal region reveals an extended helical pitch compared to the equivalent linear substrate (5.3 Å r.m.s.d. across 10 distal bp). This is consistent with the increased structural flexibility of the (−)SC substrate.

Notably, the (−)SC DNA minicircle dCas9 electron density reveals the complete path of the NTS (Extended Data Fig. 5a). Using a previous crystal structure (PDB: 5F9R) as a starting point[37], we modelled the entire NTS path from the PID, passing through the RuvC domain near the back of the HNH domain and reconnecting to the PAM distal end of the TS (Extended Data Fig. 5a–c). This model positions the NTS scissile phosphate around 17 Å away from the RuvC catalytic site (residue Ala10; Extended Data Fig. 5c). This is probably due to point mutations in the RuvC domain that disrupt metal ion coordination in the active site preventing cleavage[42]. The presence of a metal ion in the active site and allosteric activation by the HNH and L2 domains (Fig. 2a,c) would bring the NTS scissile phosphate closer. Together, these data show that Cas9 can sense different DNA topologies, regulating its catalytic activation, thereby explaining the faster cleavage rate constants observed with the (−)SC DNA substrates (Fig. 1c).

## Cas9 structure on off-target (−)SC minicircles

To determine how Cas9 accommodates mismatches in the context of (−)SC substrates, we determined the cryo-EM structure of dCas9 bound to a (−)SC minicircle encoding OT1 (Extended Data Fig. 1b). OT1 contains five mismatches throughout the protospacer, including three in the seed region, a dG–rG purine clash at position 3, three putative

wobble base pairs at positions 8, 9 and 20, and a dT–rC pyrimidine mismatch at position 15. The resulting two-dimensional (2D) class averages are similar to the diamond-ring structure of the on-target complex (Extended Data Fig. 6b). A three-dimensional (3D)-focused refinement of the core OT1 complex yields a reconstruction at 2.6 Å (Fig. 3a,b). The global architecture of the OT1 complex closely resembles the on-target complex with an r.m.s.d. of 1.6 Å (for 1041 Cα, compare Figs. 2d and 3b). However, the REC2 and HNH domains appear less-well resolved (Fig. 3a) and we therefore omit these in our structural model (Fig. 3b). Structural flexibility was observed in previously reported structures (PDB: 7QQQ and 7QQX)[8], in which both the REC2 and HNH B-factors are higher than the average for the entire structure, consistent with the idea that these domains are more dynamic.

The PID is identical to the on-target structure, and we also observe the complete 20 bp R-loop. The R-loop structure reveals all of the canonical and non-canonical base pairs across the protospacer (Fig. 3c). The seed region, considered to be the least compliant in the R-loop[43], completely accommodates all three PAM-proximal mismatches. At position 3, the dG–rG purine clash is accommodated by positioning the protospacer dG in a *syn*-conformation, resulting in a Hoogsteen base pair with two H-bonds to the guide rG facilitated by tautomerization (Fig. 3d). Notably, this base rotation is the opposite from previously reported G–G mismatches[8], in which the gRNA base adopts the *syn*-conformation to accommodate the mismatch. We propose that this strategy is facilitated by the increased flexibility of the DNA bases in the (−)SC substrate. This agrees with our previous CIRCLE-seq observation that (−)SC increases mismatch tolerance through reduced stringency of mismatch distribution and type[16]. The dC–rA mismatch at position 8 is accommodated in a typical protonated wobble configuration (Fig. 3e). The two dT–rG mismatches at positions 9 and 20 are near isosteric wobble base pairs, as expected (Fig. 3f,h). Finally, the R-loop accommodated dT–rC pyrimidine mismatch at position 15 is stabilized by a single H-bond (Fig. 3g), due to the smaller size of the bases, as described in previous off-target structures[8].

Comparing our OT1 structure to a previous precatalytic structure (PDB: 7S4U)[7], we observed that the overall Cas9 architecture remains similar, with two key differences. First, the REC2 domain is visible in 7S4U. Second, in the OT1 structure, the R-loop is completely formed, whereas the PAM-distal DNA in 7S4U is distorted upwards towards the REC3 domain with a local r.m.s.d. of around 14 Å across the TS backbone. The PAM-distal R-loop was further organized compared with PDB 7S4U (Extended Data Fig. 6e), and the REC3 domain appears more outward by around 5.4 Å (Extended Data Fig. 6i–l). Together, this off-target structure highlights the diverse strategies that Cas9 can use to accommodate a wide variety of mismatches across the protospacer on (−)SC DNA, including in the seed region. It also reveals the intrinsic flexibility of the allosterically regulated REC2 and HNH domains when bound to (−)SC off-target DNA.

To confirm that Cas9 is active with the off-target (−)SC DNA minicircles, we incubated OT1 with wild-type (WT) Cas9 in the presence of 10 mM Mg[2+] ions and froze the products onto cryo-EM grids. As for the dCas9–OT1 complex, the resulting 2D class averages exhibit a similar diamond-ring structure to the on-target complex (Extended Data Fig. 7b). Overall, the minicircle product structure is similar to a previously reported structure with a linear substrate[14]. However, a 3D-focused refinement of the core complex yields a density map at a resolution of 3.8 Å (Extended Data Fig. 7e), revealing a few key differences compared to the precatalytic complex (Fig. 3a). First, both the HNH and REC2 domains have more discernible density compared with the precatalytic OT1 structure (Fig. 3a and Extended Data Fig. 7e), indicative of a more-stable postcatalytic complex, consistent with reported product structures[7,38]. Importantly, electron density appears to be absent between the third and fourth nucleotide of the TS (from the PAM), at the expected cleavage site (Extended Data Fig. 7h). Compared with the precatalytic OT1 complex, the dG–rG mismatch also

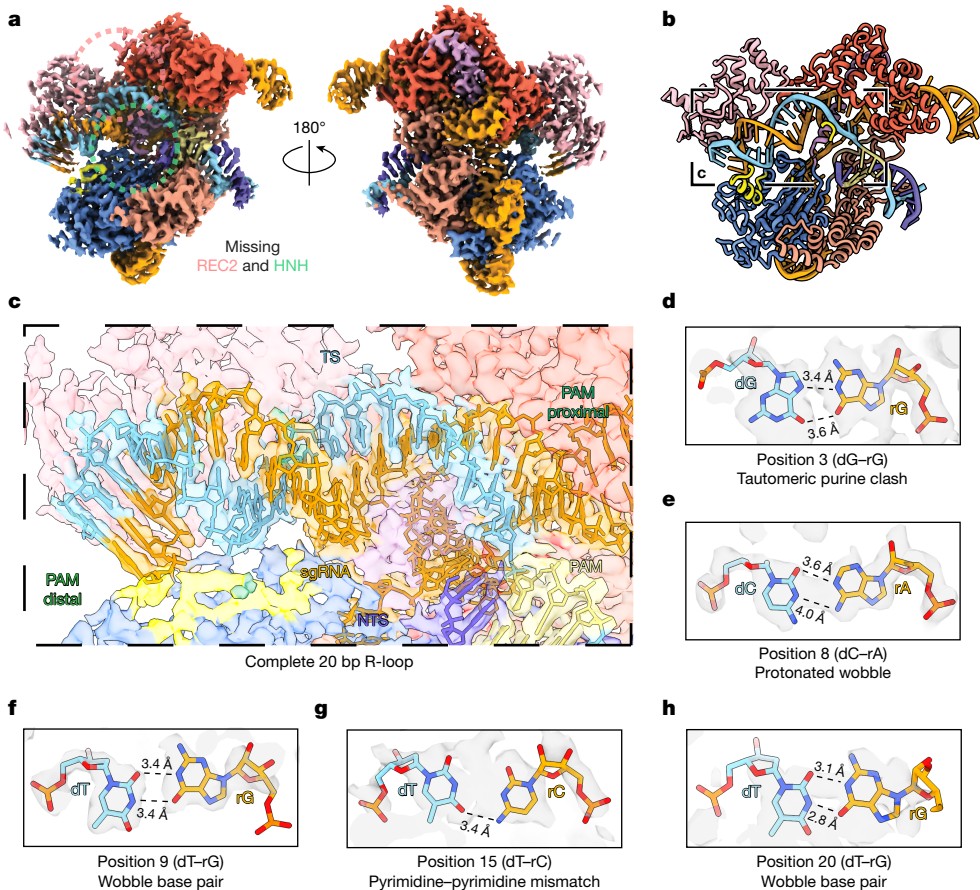

**Fig. 3 | Cryo-EM map and molecular model of the dCas9 complex with the OT1 (−)SC DNA minicircle. a**, Focus-refined cryo-EM map of dCas9−OT1 core complex (2.6 Å resolution). The dashed circles indicate flexible and unmodelled REC2 and HNH domains. **b**, Molecular model of the dCas9−OT1 complex. **c**, Cryo-EM map and molecular model of the complete 20 bp R-loop. **d**–**h**, Magnified views of mismatches at positions 3 (**d**), 8 (**e**), 9 (**f**), 15 (**g**) and 20 (**h**).

appears to be displaced (Fig. 3d and Extended Data Fig. 7i), consistent with the idea that, in the context of a (−)SC substrate, the pre-cleavage configuration is at least in part stabilized by the protospacer structure and not a random positioning of the base. Notably, the density between the second and third nucleotides is less well defined, indicating that Cas9 may be able to cleave at more than one location, consistent with previously reported alternative cleavage sites[42,44]. Moreover, some electron density of the NTS appears to be missing in the product complex indicating that this strand is more flexible, suggesting that it has been cleaved too, but this could not be observed directly.

We next determined the cryo-EM structure of dCas9 bound to a second off-target substrate, OT2 (Extended Data Fig. 1c). OT2 contains four primarily PAM-distal mismatches, including a putative dC−rA wobble base pair in the seed region (position 8), a dT−rC pyrimidine mismatch (position 15), and putative dA−rC and dT−rG wobble base pairs (positions 16 and 20). The resulting 2D class averages show distinct differences with the on-target and OT1 complexes (compare Extended Data Figs. 4b, 6b and 7b). While the entire DNA ring is still visible, it now adopts a teardrop conformation (Extended Data Fig. 7b), indicating that the DNA is not fully relaxed after dCas9 binding, consistent with partial R-loop formation. This is also shown by the AFM data, in which the dCas9-bound OT2 minicircles show higher curvature in the DNA region furthest from the protein (Extended Data Fig. 3r–t). A 3D-focused refinement of the core OT2 complex yields a density map at a resolution of 3.9 Å (Fig. 4a,b). While the global architecture of the OT2 complex resembles the on-target and OT1 complexes (compare Figs. 2d, 3b and 4b), the HNH domain appears too flexible to be resolved (Fig. 4b). Furthermore, the PAM distal end of the R-loop is also too flexible to be resolved (Fig. 4c), due to the three PAM distal mismatches. The PID is

identical to both the on-target and OT1 structures with an r.m.s.d. of 0.1 Å and 0.6 Å, respectively. However, we were only able to resolve 11 PAM-proximal base pairs in the R-loop (Fig. 4b,c). Notably, the dC−rA mismatch at position 8 is not accommodated as a wobble base pair but, rather, as Watson−Crick-like geometry (Fig. 4d). A comparison of both conformations shows that, in the wobble base pair geometry, the bases are further apart than in the tautomeric Watson−Crick-like geometry; the tautomeric Watson−Crick-like configuration is therefore more likely to occur (Extended Data Fig. 8e). A dG−rU tautomeric Watson−Crick-like structure has been previously described in the structure of an off-target complex[8], but not a dC−rA Watson−Crick-like mismatch. Notably, reorganization of a loop in the RuvC domain, previously shown to stabilize linear substrates containing PAM-distal mismatches[7], does not appear to be required with this (−)SC substrate.

The absence of discernible density in the nine PAM-distal base pairs of the R-loop raises the possibility that the increased flexibility of the (−)SC DNA bases decreases the requirement of a complete R-loop formation for cleavage activity. To test this hypothesis, we performed cleavage assays with truncated sgRNAs on (−)SC with two lengths (bp) and supercoiling densities (126 bp = −0.167, 211 bp = −0.099, respectively) and linearized DNA minicircles (Fig. 4e). Compared with the linear substrates, the data show that Cas9 cleaves (−)SC DNA minicircles with gRNAs as short as 16 nucleotides and nicks (−)SC DNA minicircles with a 14-nucleotide guide (Extended Data Fig. 8i,j) for both supercoiling densities tested (Fig. 4e). These results agree with early Cas9 studies using mismatched targets on DNA plasmids ($\sigma \approx -0.07$)[42]. These results explain the high likelihood of PAM-distal mismatches observed in the CIRCLE-seq experiments with (−)SC DNA[16], and are also consistent with our previous work showing that 14-nucleotide 'bubbles' are sufficient

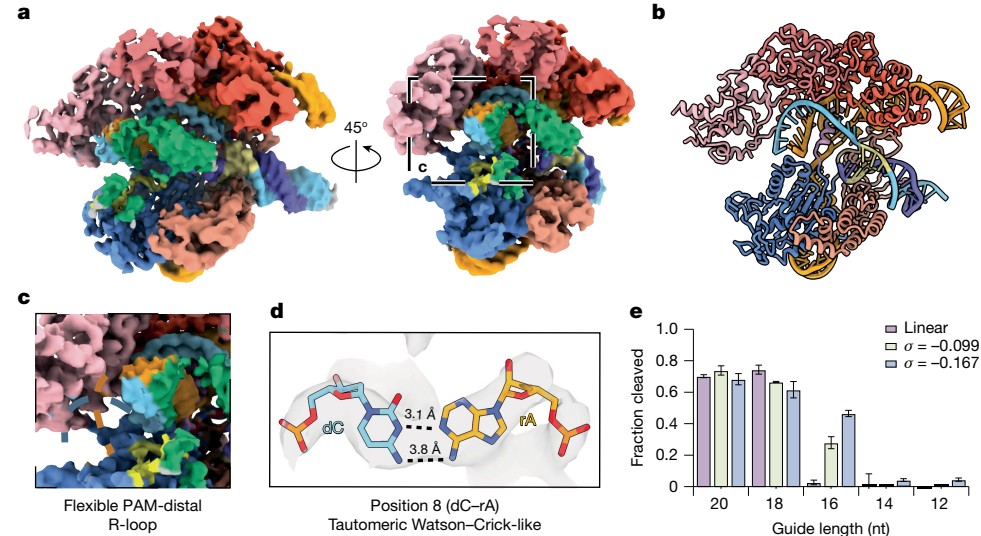

**Fig. 4 | Cryo-EM map and molecular model of dCas9 in a complex with the OT2 (−)SC DNA minicircle. a**, Focus-refined cryo-EM map of the dCas9–OT1 core complex (3.9 Å resolution), in which the HNH domain is partially visible. **b**, Structural model of the dCas9–OT2 complex, omitting the HNH domain. **c**, Magnified view of the flexible PAM-distal R-loop; a sharp decline in EM density and idealized R-loop position is denoted by the dashed blue and orange lines (for TS and gRNA, respectively) **d**, Magnified view of the dC–rA mismatch at position 8. **e**, Cleavage end points on linear (purple) or (−)SC DNA minicircles with $\sigma = -0.099$ (211 bp, green) or $\sigma = -0.167$ (126 bp, blue) with WT Cas9 and truncated gRNAs (20–12 nucleotides). Data are mean ± s.e.m. nt, nucleotide.

to promote off-target cleavage[14]. Together, the OT2 structure and characterization of truncated R-loops reveal how the flexibility of (−)SC minicircles considerably decreases the requirement for PAM distal base-pairing to activate Cas9 cleavage and explains how PAM distal mismatches in the protospacer are readily tolerated.

## HNH dynamics on off-target (−)SC DNA

Compared with the on-target structure, the HNH domain in both (−)SC off-target structures appears to be more flexible (Figs. 3a and 4a). To confirm that binding to (−)SC off-target substrates results in a more flexible HNH domain, we used single-molecule FRET (smFRET). Single-molecule approaches eliminate ensemble averaging and, therefore, enable imaging of conformational dynamics in real time. To this aim, we doubly labelled the HNH (Cys867) and REC1 (Cys355) domains of dCas9 with a FRET pair (Fig. 5a and Extended Data Fig. 9a), as described previously[12], and negatively supercoiled a torsionally constrained λ-DNA substrate using optical tweezers (Fig. 5b) as described previously[16,45]. We determine the supercoiling density by measuring the distance increase in the force-extension curve at 70 pN (Fig. 5c).

At low force (5 pN) and with relaxed DNA ($\sigma = 0$), the resulting kymographs reveal a single on-target binding event (Fig. 5d), as expected. In agreement with our on-target structure and previous smFRET data[12], the bound dCas9 complex exhibits a clear high (-0.7) FRET ratio (Fig. 5d), indicating that the HNH and the REC1 domains are in close proximity. When the tethered DNA is (−)SC ($\sigma \approx -0.2$), we can still observe on-target bound complexes (Extended Data Fig. 9c), albeit with a higher FRET ratio (around 0.8–1), indicating that the HNH domain is closer to the REC1 domain, agreeing with the 15 Å swing observed in our on-target structure (Fig. 2). Similar to our previous high-force experiments[22,24], we also observed many off-target events with similar HNH dynamics (Extended Data Fig. 10j–l). Moreover, we observed numerous transient off-target binding events (Extended Data Fig. 9b,c). The majority (87%, $n = 125$) of these dissociate rapidly and exhibit no FRET ($t = 1.6 \pm 0.2$ s; Extended Data Fig. 9g). These transient no-FRET binding events probably correspond to complexes that have not successfully formed the R-loop, but also include molecules with a photobleached acceptor or two donors. The remaining off-target binding events (13%, $n = 19$) exhibit higher FRET ratios (ranging from 0.1 to 0.6;

Extended Data Fig. 9i), with dynamic transitions between both low and high FRET confirmations (Fig. 5e,f) and threefold slower dissociation times ($t = 4.9 \pm 2.0$ s; Extended Data Fig. 9h). Notably, molecules with higher FRET ratios tend to bind longer (Extended Data Fig. 9j). These data suggest that off-target-bound dCas9 molecules are stabilized by partial R-loop formation (longer bound times) with the HNH and REC2 domains in closer proximity (higher FRET). Together, these results confirm that HNH domain flexibility and proximity to REC1 are not only sensitive to the presence of mismatches but are also allosterically regulated by the topological state of the DNA substrate, which in turn accelerates off-target cleavage. Akin to previously described conformational checkpoints in the reaction pathway[40], we propose that (−)SC reduces the potential energy barriers to unwind the DNA substrate (dsDNA) and facilitates R-loop formation, even in the presence of multiple mismatches. Once the R-loop is formed, allosteric activation (REC2 and HNH rearrangement) can occur, leading to fast DNA cleavage.

Numerous high-fidelity variants have been developed to reduce Cas9 off-target activity. We sought to test two of the latest ones (SuperFi and Sniper2L Cas9)[7,27] using our optical tweezers with (−)SC λ-DNA (Extended Data Fig. 10b,c). In the presence of calcium ($Ca^{2+}$) ions to prevent cleavage, the resulting kymographs show that both SuperFi and Sniper2L Cas9 bind to numerous off target sites on the (−)SC DNA, even at low force (5 pN) and comparable to Cas9 with similar average bound times ($4 \pm 1$ s compared with $5 \pm 1$ s for WT Cas9). We also tested whether these two variants can cleave off-target sequences in the context of our (−)SC minicircles. The cleavage assays show that Sniper2L Cas9 cleaves both OT1 and OT2, whereas SuperFi Cas9 cleaves OT1 and nicks OT2 (Extended Data Fig. 10e). Similarly, to WT Cas9, neither high-fidelity variant cleaves linearized OT1 or OT2 (Extended Data Fig. 10f).

## Discussion

CRISPR–Cas9 is a versatile tool for targeted genome editing[46,47]. However, spurious off-target events limit its safe use in therapeutic applications. Understanding the molecular basis of off-target activity has led to the development of various high-fidelity Cas9 nucleases that reduce off-target effects[7,27,48,49]. Magnetic and optical tweezer studies have shown the importance of supercoiling in R-loop formation[3,15],

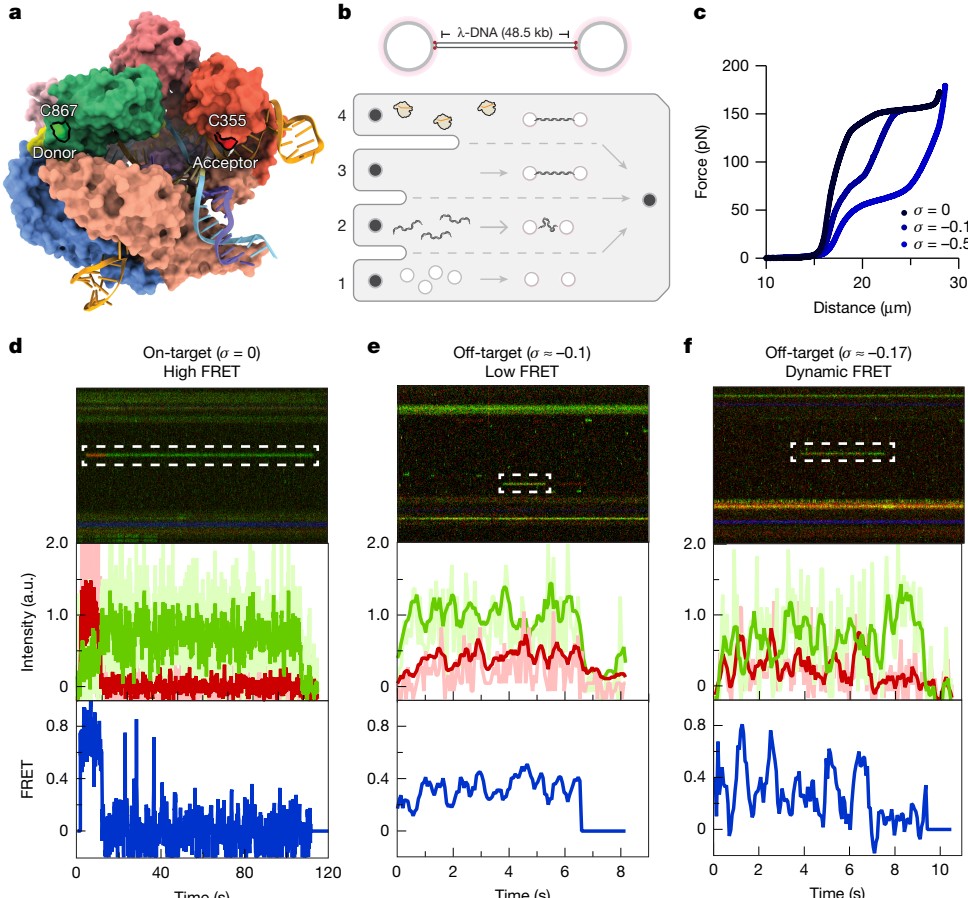

**Fig. 5 | Single-molecule optical-tweezer and FRET analyses show on- and off-target HNH conformational dynamics. a**, Surface model of the on-target dCas9 core complex, highlighting residues Ser355 and Ser867, which were converted to cysteine residues and dual labelled with the FRET pair Cy3/Cy5, as described previously[18]. **b**, Schematic of experimental setup. Top, optically trapped (pink) streptavidin-coated beads and torsionally constrained λ-DNA; bottom, the microfluidic set-up: (1) bead channel, (2) DNA channel, (3) buffer channel and (4) protein and imaging channel. **c**, Force–extension curve of torsionally constrained λ-DNA under increasing supercoiling densities ($\sigma = 0$ to $-0.5$). **d**, Representative on-target kymograph with Cas9 bound in a high-FRET conformation, at 5 pN, $\sigma = 0$; the corresponding intensities and FRET trajectories are shown below. **e**, Representative off-target kymograph of Cas9 bound in a low-FRET conformation, at 5 pN, $\sigma \approx -0.1$; the corresponding intensities and FRET trajectories are shown below. **f**, Representative off-target kymograph with dynamic transitions between high- and low-FRET states, at 5 pN, $\sigma \approx -0.17$; the corresponding intensities and FRET trajectories are shown below. a.u., arbitrary units.

and that both DNA stretching and (−)SC can induce Cas9 off-target activity[14,16].

Recent Cas9 structures bound to linear off-target sequences have revealed that R-loop formation is enabled by non-canonical base-pairing between the guide and the target DNA[8]. Linear substrates containing PAM-distal mismatches can also be stabilized by reorganization in the RuvC domain[7]. However, the molecular mechanism behind topology-induced off-target cleavage has remained elusive.

To address this, we developed small programmable (−)SC minicircles as bona fide Cas9 substrates. Using high-resolution AFM, we show that supercoiling the minicircles changes their relaxed circular conformation to collapsed, or double-denatured, consistent with previous MD simulations[34]. Cas9 binding to (−)SC minicircles induces an open diamond-ring-like structure (Fig. 1g) by local topology relaxation through the change of twist ($\Delta Lk = -2$) expected for complete R-loop formation[3,15].

Using cryo-EM, we determined high-resolution features of the dCas9 complex. The density confirms the diamond-ring-like structure. Globally, the Cas9 complex adopts a similar bilobed conformation to that on linear substrates[38]. However, on the (−)SC minicircle, the HNH domain swings 15 Å closer to the TS scissile phosphate, poising the complex closer towards catalysis (Fig. 2g,h). A previous structure of dCas9 bound to linear DNA shows a flexible, lower-resolution HNH domain[50]. In contrast to previous studies[7], we observe the entire NTS

path, from PID through the RuvC domain and to the PAM distal dsDNA duplex (Extended Data Fig. 5a–d).

Earlier structures of Cas9 with partial dsDNA[37], PAMer (lacking a NTS, Extended Data Fig. 5h)[36] and ssDNA[35] targets show similar overall Cas9 architectures and conformations of the HNH domain, but the REC2 and REC3 domains appear to be more outward than in our structure (Extended Data Fig. 5f–i), consistent with the idea of a more relaxed R-loop checkpoint in the supercoiled structure. Furthermore, we have previously shown that bubble and PAMer substrates cleave off-target sequences faster than double-stranded linear substrates[14].

Taken together, these observations indicate that Cas9 can sense the substrate topology, which in turn regulates HNH domain conformation, thereby providing a reasonable explanation for the faster cleavage rates observed in bulk cleavage assays with (−)SC DNA compared with linear substrates[8]. As described in MD simulations[41], concomitant dynamics of the REC2 and HNH domains (Supplementary Video 2) indicate that allosteric activation is preserved in the (−)SC substrate. Consistent with our structures (Figs. 2–4) and previous studies[9,10,12], our smFRET data reveal HNH domain dynamics on off-target sequences (Fig. 5f). In the context of the Cas9 energy landscape[40], (−)SC lowers the transition-state barrier for R-loop formation, thereby accelerating cleavage[3,15].

We also obtained two (−)SC off-target structures (Figs. 3 and 4) demonstrating how diverse mismatches are accommodated across the

protospacer, including in the seed region. These reveal non-canonical base-pairing geometries, such as tautomeric Watson–Crick-like base pairs and tautomeric purine clashes (Figs. 3d and 4d), consistent with CIRCLE-seq data with (−)SC substrates[16]. Compared with previous off-target structures[8], more flexible (−)SC substrates may facilitate sterically challenging mismatches in the R-loop. These differences could arise from various mismatch types and the topological context (for example, (−)SC DNA versus PAMer substrate lacking the NTS). However, the REC3 domain appears displaced by around 5.4 Å in both OT1 and OT2 structures, which may increase the tolerance for PAM-distal mismatches.

Although both on-target and OT1 minicircles form complete R-loops and are cleaved by Cas9, it may seem surprising that the HNH domain is better resolved in the on-target structure. However, the increased mobility of the HNH and REC2 domains in OT1 is consistent with allosteric activation, as described previously[11,41,51,52], and with the HNH domain smFRET off-target dynamics (Fig. 5f).

Our post-cleavage structure highlights how (−)SC substrates facilitate Cas9 cleavage even in the presence of PAM-proximal mismatches. The Cas9 product complex maintains the diamond-ring-like structure, in agreement with our previous tweezer experiments[14] and in vivo studies showing that Cas9 must be actively removed by replication or transcription[53,54], but in contrast to previous AFM studies[55]. The product structure (Extended Data Fig. 7h) also reveals a second possible cleavage site, consistent with staggered Cas9 cleavage[42].

PAM-distal mismatches in previous studies with linear substrates appeared fully resolved[8] and stabilized by RuvC domain conformational changes[7]. However, in (−)SC substrates, the OT2 structure (Fig. 4) shows that PAM-distal mismatches do not need to be accommodated into the R-loop to maintain activity, revealing a reduced base-pairing requirement beyond the seed region. Previous studies have suggested that at least 15 bp is required for activity[56]. Our data are in agreement, with (−)SC minicircles pointing at a topology-dependent HNH activation checkpoint (Fig. 4e and Extended Data Figs. 8i,j and 10g–i). This poses implications for using truncated guides to increase specificity. Specific-cell-cycle-stage editing, for example, using very fast CRISPR (vfCRISPR)[57], may further reduce off-target effects by avoiding topology-influencing processes.

A potential limitation of our minicircles is that the supercoiling densities used here ($\sigma = -0.167$ and $-0.099$) may not reflect the average state of the human genome. However, psoralen binding studies in vitro[58] and in vivo[18] have estimated a supercoiling density of $\sigma \approx -0.06$, showing that our data at $\sigma = -0.099$ approach physiological supercoiling levels. Further investigation of genome-wide off-target events across different supercoiling states would be interesting.

In summary, we present insights into the basis of Cas9 targeting and conformational activation in the context of (−)SC substrates. (−)SC enables the formation of previously unobserved non-canonical base-pairing geometries that facilitate R-loop formation, while maintaining dynamic allosteric activation for cleavage. By understanding the effect of (−)SC on Cas9, these studies will help design the next-generation of high-fidelity CRISPR effectors. Furthermore, these (−)SC minicircles provide an approach to investigate the role of DNA topology on DNA-binding proteins.

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

## Methods

### DNA and RNA preparation

All DNA oligonucleotides were ordered from IDT (Supplementary Table 1). DNA oligos were resuspended to 200 µM and subjected to 15% denaturing PAGE, extracted from the gel through ultraviolet shadowing, crushed and soaked, ethanol precipitated, quantified by spectrophotometry (Biodrop) and stored at −20 °C. Some DNA oligos were labelled (Supplementary Table 1) using NHS-ester labelling chemistry, as described previously[59], and purified by high-performance liquid chromatography.

### In vitro transcription of sgRNA

DNA templates required for sgRNA were annealed together under equimolar conditions by heating to 95 °C for 2 min before cooling by −2 °C min⁻¹ to 4 °C in an annealing buffer containing 50 mM Tris-HCl pH 8 and 100 mM NaCl. The annealed dsDNA template was subjected to in vitro transcription containing 500 mM template, 5 mM of each dNTP, 25 mM MgCl₂, 0.2 U of *Escherichia coli* inorganic pyrophosphatase, 80 U of RNase inhibitor, 7.5 mM DTT, 1× RNA polymerase buffer and 750 U of RNA polymerase. This reaction was mixed well and incubated at 37 °C overnight. The reaction was subjected to 15% denaturing PAGE and then purified by ethanol precipitation.

### Protein expression, purification and fluorescence labelling

WT and dCas9 were gifted by E. Gordon, and they were expressed and purified as described previously[16]. Dual-labelled Cas9/dCas9 was labelled with a 1:20:40 ratio of Cas9/dCas9:Cy3 maleimide:Cy5 maleimide in Cas9 storage buffer (20 mM HEPES pH 7.5, 500 mM KCl, 1 mM TCEP). The reaction was left for 2 h at room temperature and then overnight at 4 °C in the dark before loading onto the Superdex 200 Increase 10/300 GL size-exclusion column. The labelled fractions were pooled, quantified using the Biodrop spectrophotometer, aliquoted, flash-frozen in liquid nitrogen and stored at −80 °C.

### DNA minicircle assembly

Minicircles were assembled using an adapted protocol described previously[60]. The design of the minicircles included several AT-tracts to promote bendability, as described[60–63]. Three main steps were used: first, the ssDNA (126 nucleotides) and splint DNA (29 nucleotides) were heat-annealed together at 95 °C for 3 min and subsequently cooled down on ice for 1 min. The reaction was incubated with T4 DNA ligase (20 U) at 37 °C for 30 min to form a ssDNA minicircle. Secondly, the ssDNA minicircle was subjected to T4 DNA polymerase fill-in, the ssDNA minicircle was diluted into a reaction containing T4 DNA polymerase (120 U) and T4 DNA ligase (16,000 U) in the presence of 100 µM dNTP mix and 10 mM ATP, and the reaction was incubated at 12 °C for 1 h. Finally, the reaction was treated with Exo V (20 U) and T5 Exo (20 U) in the presence of 1 mM ATP at 37 °C for 45 min; this was done to remove all incompletely assembled minicircles. To clean the reaction further, it was treated with proteinase K to remove any remaining protein in the reaction before passing the reaction through a Monarch PCR & DNA Clean-up Kit (NEB).

### Negative supercoiling of DNA minicircles

Minicircles were subjected to negative supercoiling using *E. coli* gyrase (Inspiralis) in the presence of EtBr. A reaction of 10 nM minicircle DNA (mcDNA), 1× gyrase buffer, 0.1 mg ml⁻¹ EtBr and 20 U µl⁻¹ of *E. coli* gyrase was incubated at 37 °C for ≥1 h. The reaction was then cleaned either by phenol–chloroform extraction followed with ethanol precipitation (AFM) or using the Monarch PCR & DNA Clean-up Kit (Bulk/Cryo-EM).

### Bulk cleavage assays and kinetics

DNA minicircles were processed for bulk cleavage assays. Cas9 was first complexed at 1 µM with a 1:1 ratio of Cas9 to sgRNA at room temperature for 10 min. The complexed Cas9 (now termed RNP) was mixed in a 1:10 ratio of mcDNA to RNP and incubated at 37 °C for 20 min in a cleavage buffer (100 mM NaCl, 50 mM Tris-HCl pH 8, 10 mM MgCl₂). Timepoints were taken at 0 s, 20 s, 40 s, 1 min, 2 min, 5 min, 10 min and 20 min intervals. The reactions were stopped by adding equal parts of Cas9 STEB (100 mM Tris-HCl pH 8, 100 mM EDTA, 0.2% SDS, 40% sucrose, 2 mg ml⁻¹ RNase A). The stopped reactions were briefly vortexed and centrifuged to be loaded onto 10% native PAGE (with 10 mM CaCl₂). Gels were analysed on Fiji (ImageJ), and intensities for SC, linear and background bands were measured. All uncropped gels are located in Supplementary Fig. 1.

The fraction cleaved was defined by the following equation:

$$\frac{Lin}{Lin + Nick + Sc}\text{,}$$

where intensities are background subtracted. Kinetics were fitted to a single exponential using IGOR Pro (Wavemetrics).

### AFM sample preparation and imaging

DNA minicircles were adsorbed onto freshly cleaved mica discs (diameter approximately 5 mm) using a divalent cation protocol fully described in a published protocol[64]. An aqueous nickel buffer solution (20 µl, 3 mM NiCl₂, 20 mM HEPES, pH 7.4) was placed onto the mica discs and minicircle DNA samples (2–5 ng) pipetted into the meniscus formed by the buffer. After incubation at room temperature for 30 min, the unbound DNA was removed by washing the mica disc with four 20 µl portions of the nickel buffer. A further 20 µl of buffer was added to the sample for imaging.

The dCas9–sgRNA complex (2 ng) was incubated with DNA minicircles in a 1:1 ratio at a concentration of 2 ng µl⁻¹ in 1× NTM and pre-incubated at room temperature for 10 min. These samples were then immobilized as described above using nickel buffer solution (20 µl, 3 mM NiCl₂, 20 mM HEPES, pH 7.4). AFM imaging was performed using a FastScan Dimension XR microscope (Bruker) and FastScan-D AFM probes (Bruker) in PeakForce tapping mode. Force–distance curves were recorded at a frequency of 8 kHz using with a PeakForce amplitude of 10 nm. The PeakForce setpoint, which controls the tip–sample interaction force, was set in the range of 7–16 mV as referenced from the force baseline resulting in peak sample interaction forces of 50–120 pN.

### Automated flattening and analysis of AFM images

AFM images were processed using the open-source AFM image analysis software TopoStats[33,64]. TopoStats used median row alignment, planar tilt removal, quadratic tilt removal and scar removal to flatten the images. A background mask obtained from pixel heights below $1\sigma$ height was used to repeat the steps on just the background data to improve flattening accuracy. The image data were translated so the background average centred around 0. Finally, a 1.1 px Gaussian filter was applied to reduce high-gain noise.

### Molecular quantification from AFM images

Grain detection and analysis was performed using Python notebooks to derive structural information such as the minimum width (Feret diameter), contour length and aspect ratio. First, grains were detected using a generous threshold, then vetted for viability based on size. Grains were then remasked more accurately and a trace was produced to approximate the path of the DNA backbone. From these, statistics were calculated such as minimum width (Feret diameter), contour length and aspect ratio (Extended Data Fig. 2). Central mean curvature analysis was performed by measuring the curvature of the DNA trace in the centre at the point furthest from the bound dCas9. The midpoint of the trace was determined automatically, and the curvature was calculated 5 nm either side of the midpoint. The mean curvature was then calculated over that 10 nm segment for each molecule. Differences between the

two distributions were measured using a *t*-test. The difference was found to be significant, with a *P* value of 0.033.

## Cryo-EM grid preparation and data collection

dCas9 or wtCas9 RNP was complexed with negatively supercoiled mcDNA at room temperature for 10 min in cleavage buffer (100 mM NaCl, 50 mM Tris-HCl pH 8, 10 mM $MgCl_2$) in a 1:1 stoichiometric ratio at 250 nM each. Then, 3.5 µl of 250 nM dCas9 or Cas9(WT)–mcDNA complex was loaded onto glow-discharged copper Quantifoil R2/2 grids with additional 3 nm thick carbon support; they were blotted using a FEI Vitrobot Mark IV for 0.5–1 s in a controlled room at 4 °C and 100% humidity and then plunged into liquid ethane kept at liquid nitrogen temperatures. Grids were first screened on a Thermo Fisher Scientific Talos F200i electron microscope at 200 kV before high-resolution data collection on the Thermo Fisher Scientific Titan Krios electron microscope at 300 kV acceleration voltage equipped with a K3 direct electron detector. The defocus range was set from −2.5 µm to −0.5 µm.

## Cryo-EM data processing

The collected datasets were processed using RELION4[65], micrographs were CTF- and motion-corrected[66,67]. Micrographs with estimated resolutions worse than 7 Å from CTF fitting were removed as a starting point for processing. On-the-fly processing was performed where particles were picked automatically with Topaz[68] and 2D classification was performed. The best classes were subset-selected and used as a reference for reference-based picking. Particles were picked and underwent several rounds of 2D and 3D classifications before a final 3D refinement. For focus refinement maps, reference maps were subjected to a soft mask creation in RELION, masks were used to remove noise from the remaining minicircle DNA.

## Cryo-EM modelling

An unpublished model based on PDB 6O0Z was used as a starting model. An AlphaFold3 model of apo *Sp*Cas9 was used to fill in regions that were previously unmodelled, including residues 512–573, 611–678 and 685–730. Modelling was performed in Coot[69]; first, the coordinates were rigid-body fitted to each map using ChimeraX[70], after minimization using Namdinator[71], and refined further in Coot by real space refinement, manually updating the positioning and ID of each residue, DNA and RNA nucleotide. Finally, the initial models were used as a reference for real-space refinement in Phenix with three macro cycles under the default reference restraints.

## Optical tweezer correlated fluorescence microscopy experiments and analysis

Using the commercially available LUMICKS C-trap, we capture and induce (−)SC on torsionally constrained λ-DNA using optical tweezers as described previously[45]. After cleaning the system according to the manufacturer's instructions, channels were passivated in 0.5% Pluronics PBS, protein channels were additionally passivated using 0.2 mg ml$^{-1}$ BSA.

Two 4.35 µm SPHERO streptavidin-coated polystyrene particles (0.005%, w/v) were optically trapped using infrared lasers. Subsequently, torsionally constrained λ-DNA (48.5 kb) was captured and suspended between the two beads under flow. The tethered DNA was subjected to a force–distance curve to characterize its integrity, topology (unconstrained versus torsionally constrained) and overstretching characteristics. DNA was negatively supercoiled using optical tweezers (σ, −0 to −0.5) and 1 nM dCas9 cy3/cy5 RNP was flowed in the protein channel to capture binding using confocal imaging. RNPs used contain a guide targeting the λ2 site on λ-DNA as described previously[9,14,16].

All data collected were extracted using LUMICKS Lakeview. Force–extension curves were exported in the CSV format, and replotted in IGOR Pro. Kymographs were exported in the TIFF format, and the green and red channels were isolated to extract intensities for FRET analysis. Cas9 off-target events were quantified using LUMICKS Cas9 kymotracker scripts (https://github.com/lumicks/harbor/tree/main/Analysis/(LUMICKS)%20Cas9%20binding%20to%20DNA%20using%20the%20kymotracker).

## FRET analysis

Molecules that exhibited FRET dynamics were isolated in Fiji, a nine-pixel vertical window around the trajectory was made and the average intensities were obtained. Intensities were manually background-subtracted by generating an average intensity at the end of the trajectory and subtracting this value from averaged intensities. Background-subtracted intensities were plotted and subsequently used for FRET values. Green and red intensities were smoothed to reduce noise throughout the traces using IGOR Pro smoothing function with a box algorithm and a five-pixel smoothing window.

FRET was calculated as:

$$\frac{I_A}{I_A + I_D},$$

where $I_A$ is the background subtracted intensity of the acceptor and $I_D$ is the background subtracted intensity of the donor.

## Reporting summary

Further information on research design is available in the Nature Portfolio Reporting Summary linked to this article.

## Data availability

Electron density maps have been deposited at the Electron Microscopy Database (diamond ring on-target dCas9 complex, EMD-51860; core on-target dCas9 complex, EMD-51861; core OT1–dCas9 complex, EMD-51862; core OT2–dCas9 complex, EMD-51863; and core OT1–Cas9(WT) complex, EMD-51864) and atomic coordinates have been deposited at the Protein Data Bank (core on-target dCas9 complex, 9H4J; core OT1–dCas9 complex, 9H4K; core OT2–dCas9 complex, 9H4L; and core OT1–Cas9(WT) complex, 9H4M). All unique materials are available on request after completion of a standard materials transfer agreement. Source data are provided with this paper.

## Code availability

Code for the AFM post-processing steps and analyses is available at GitHub (https://github.com/AFM-SPM/TopoStats/tree/SylviaWhittle/hariborings).

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

**Acknowledgements** We thank the current and former members of the Rueda laboratory for comments and suggestions; E. Gordon for the initial provision of purified Cas9 protein and N. Cronin for technical assistance on data acquisition, preprocessing and data storage. We thank D. Wigley for financially supporting A.S.B.J. We acknowledge the support of the Henry Royce Institute for Advanced Materials for Q.M.S. through the Student Equipment Access Scheme, enabling access to Bruker Dimension XR facilities at The Royce Discovery Centre at the University of Sheffield; EPSRC grant numbers EP/R00661X/1 and EP/P02470X/1. The Rueda laboratory is supported by a core grant of the MRC-Laboratory of Medical Sciences (UKRI MC-A658-5TY10). The Pyne laboratory is supported by UKRI Future Leaders Fellowship (MR/W00738X/1) and the Medical Research Council (MR/W006944/1).

**Author contributions** Conceptualization: Q.M.S., A.L.B.P. and D.S.R. Methodology: Q.M.S., R.J.A., S.W., D.E.R. and D.I.E. Investigation: Q.M.S., R.J.A., S.W., A.S.B.J., D.E.R. and D.I.E. Visualization: Q.M.S., S.W., A.S.B.J., D.E.R., A.L.B.P. and D.S.R. Funding acquisition: A.L.B.P. and D.S.R. Project administration: Q.M.S., A.L.B.P. and D.S.R. Supervision: A.L.B.P. and D.S.R. Writing—original draft: Q.M.S. and D.S.R. Writing—review and editing: Q.M.S., R.J.A., S.W., D.E.R., A.S.B.J., D.I.E., K.L.M., A.L.B.P. and D.S.R.

**Additional information**
**Correspondence and requests for materials** should be addressed to Alice L. B. Pyne or David S. Rueda.

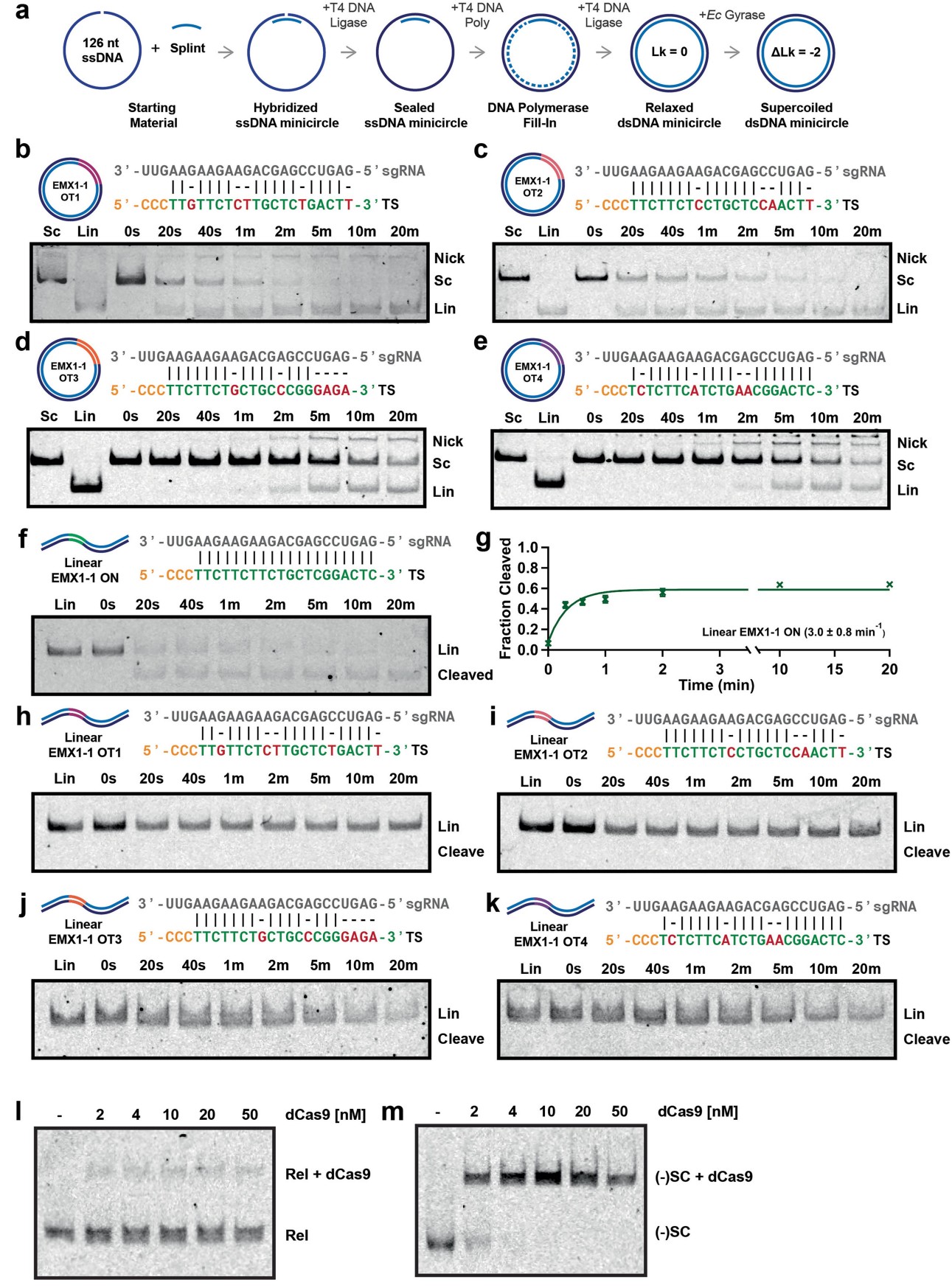

**Extended Data Fig. 1** | See next page for caption.

**Extended Data Fig. 1 | Synthesis and characterization of negatively supercoiled minicircle DNA. a**, Assembly scheme of (−)SC minicircle DNA. **b**−**e**, Representative native PAGE of EMX1-1 OT1-OT4 with sequences (top) and EMX1-1 OT1 contains 5 mismatches across the protospacer – top, cleavage time course of OT1-4 with WT Cas9 (bottom). **f**, Sequence of EMX1-1 ON DNA minicircle (top) and representative Native PAGE of linear cleavage time course with WT Cas9. **g**, Cleavage kinetics of on-target linear time course. **h**−**k**, Sequence of EMX1-1 OT1-OT4 target site (top) and representative native PAGE of linear cleavage time course with WT Cas9. **l**−**m**, EMSAs with WT Cas9 versus relaxed DNA minicircles and (−)SC DNA minicircles respectively.

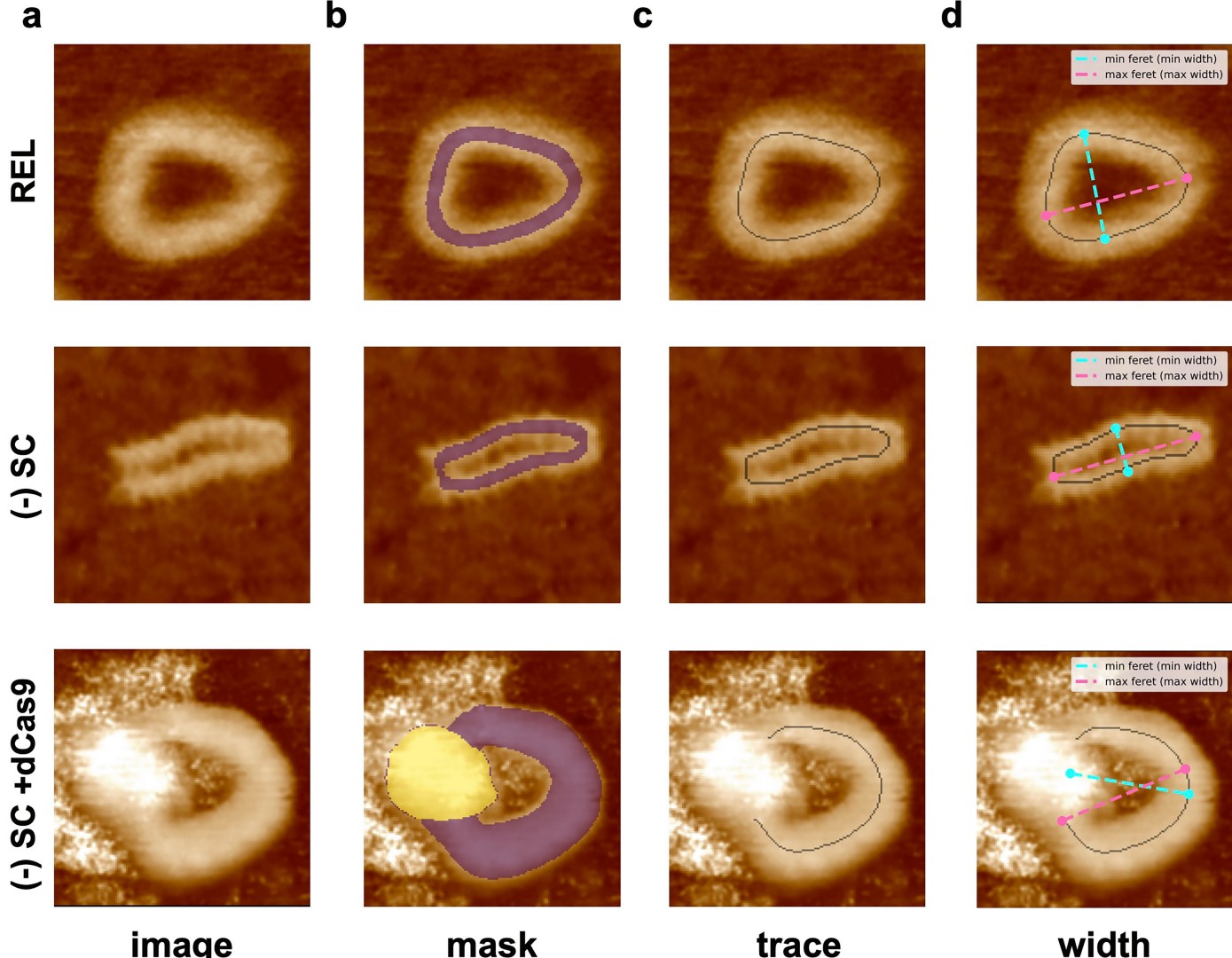

**a** **b** **c** **d**

REL

(-) SC

(-) SC +dCas9

**image** **mask** **trace** **width**

**Extended Data Fig. 2 | Segmentation and skeletonization pipeline for individual molecules from Atomic Force Microscopy images.** Images are segmented to form a bounding box around an individual molecule **a**, to create a binary mask of an individual molecule **b**, skeletonized to a single pixel trace along the backbone of the on-target DNA molecule **c**, and the minimum width and maximum length calculated from the skeletonized trace **d**, For molecules with Cas9 bound two separate binary masks were created, one for the protein which was not skeletonized and one for the on-target DNA which was skeletonized.

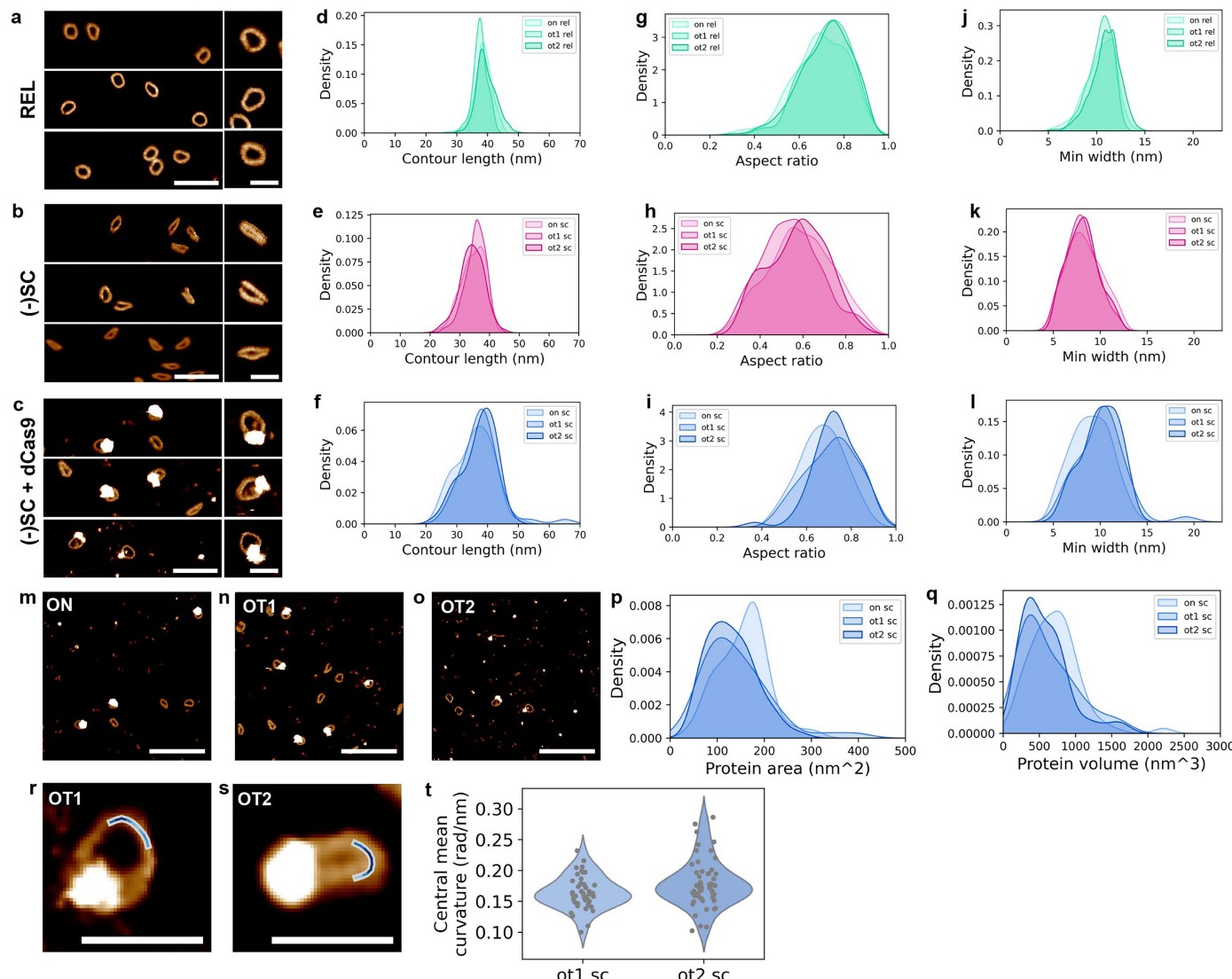

**Extended Data Fig. 3 | Quantification of structural changes in minicircles in response to supercoiling and binding of dCas9 by Atomic Force Microscopy.** Images of DNA minicircles with on-target, off-target 1 and off-target 2 sequences in relaxed **a**, and supercoiled **b**, forms, and in supercoiled form with dCas9 bound **c**. Quantification of DNA minicircle structure for on-target, off-target 1 and off-target 2, by contour length (**d–f**), aspect ratio (**g–i**) and minimum width for each molecule (**j–l**). Scale bars = 50 nm (wide view images), 20 nm (single images). Representative micrographs of on-target, off-target 1 and off-target 2 (−)SC DNA minicircle-dCas9 complexes respectively (**m–o**), Probability density of protein area (nm$^2$) **p**, and protein volume (nm$^3$) **q**, for analysed molecules. Micrographs showing OT1 and OT2 (−)SC DNA minicircle-dCas9 complexes with the highest central mean curvature (intensity of curvature overlaid) from each distribution (**r–s**), scale bars = 20 nm. Probability density of central mean curvature (rad/nm) **t**, for OT1 and OT2 (−)SC DNA minicircle-dCas9 complexes.

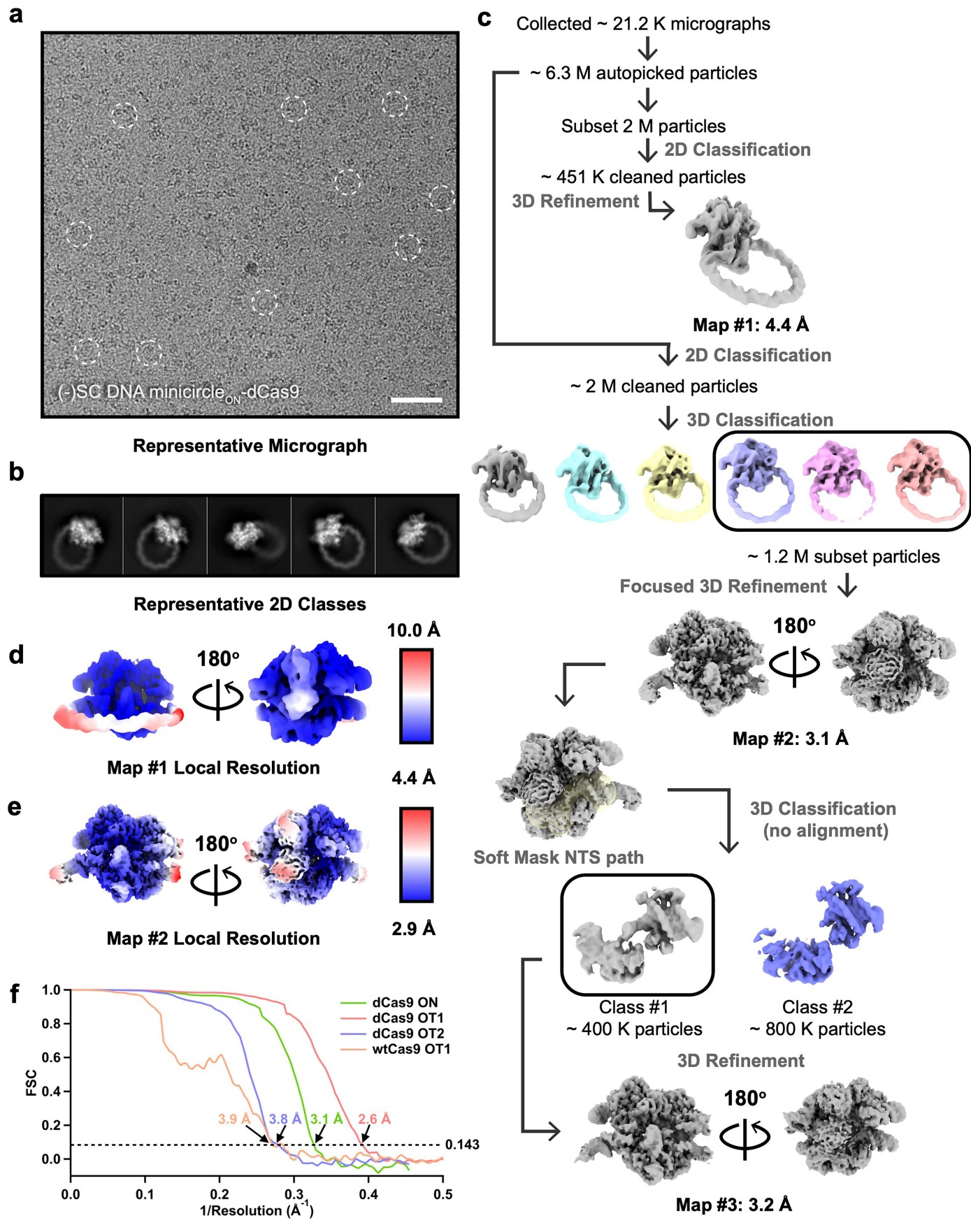

**Extended Data Fig. 4 | CryoEM data processing pipeline of on-target (−)SC DNA minicircle complex. a**, Representative micrograph of on-target (−)SC DNA minicircle complex. **b**, Representative 2D classes, showing 'diamond ring' structure. **c**, Workflow in RELION, showing micrographs collected, through 2D classification and final high-resolution 3D reconstructions. **d**, Local resolution of Map #1. **e**, Local resolution of Map #2. **f**, Fourier Shell Correlation Curves for all focus-refinement maps in this study.

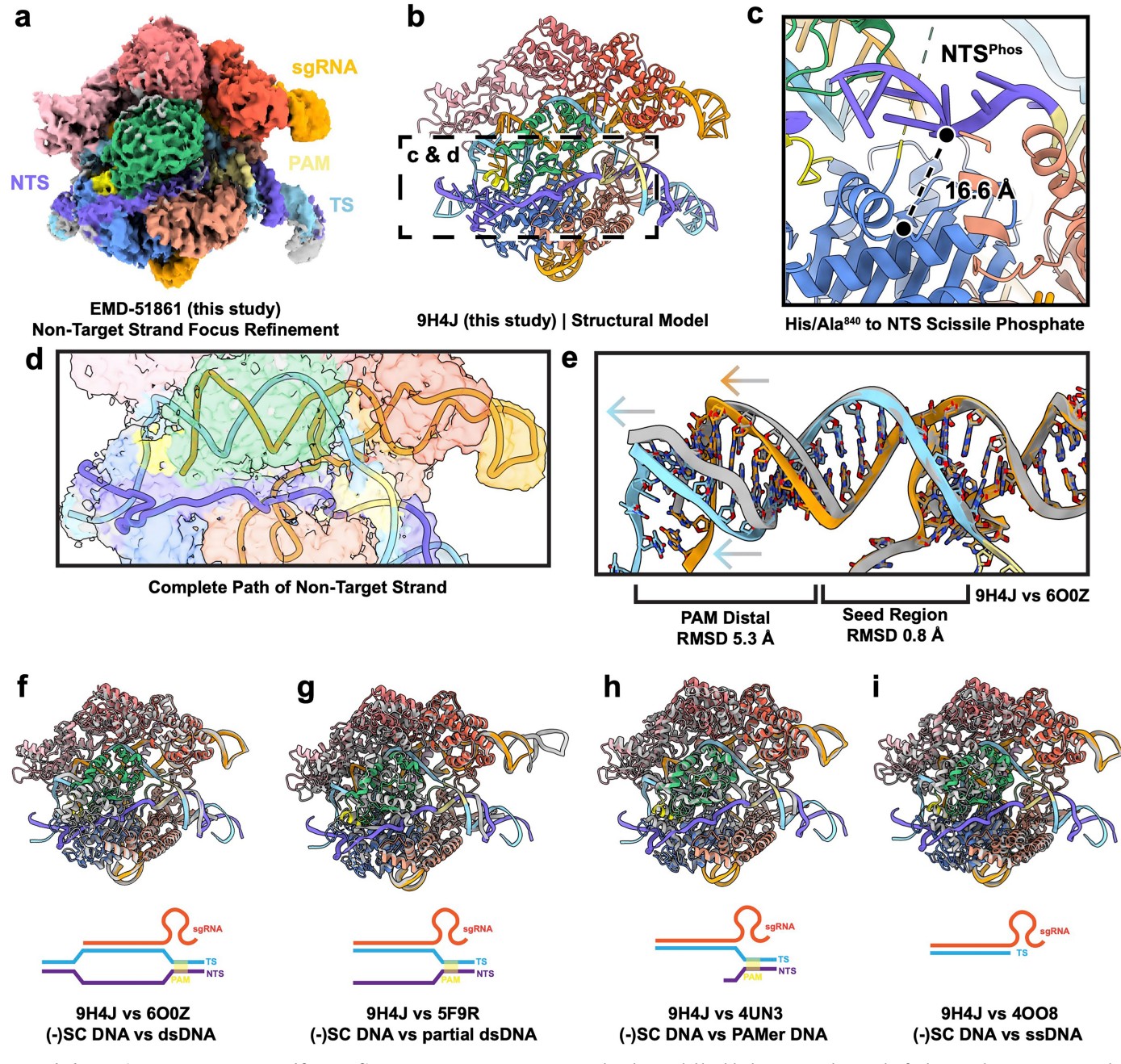

**a** EMD-51861 (this study)
Non-Target Strand Focus Refinement

**b** 9H4J (this study) | Structural Model

**c** His/Ala[840] to NTS Scissile Phosphate

NTS[Phos]

16.6 Å

**d** Complete Path of Non-Target Strand

**e** 9H4J vs 6O0Z

PAM Distal RMSD 5.3 Å    Seed Region RMSD 0.8 Å

**f** 9H4J vs 6O0Z
(-)SC DNA vs dsDNA

**g** 9H4J vs 5F9R
(-)SC DNA vs partial dsDNA

**h** 9H4J vs 4UN3
(-)SC DNA vs PAMer DNA

**i** 9H4J vs 4OO8
(-)SC DNA vs ssDNA

**Extended Data Fig. 5 | Non-target strand focus refinement on on-target (−)SC DNA minicircle complex. a**, Focus-refined CryoEM map of dCas9-ON core complex (3.1 Å resolution) including NTS (Map #3 – EMD-51861). **b**, Molecular model of Map #3 (PDB – 9H4J). **c**, Distance map of the catalytic residue D10A to the non-target strand scissile phosphate. **d**, CryoEM map and molecular model highlighting complete path of R-loop and NTS. **e**, Structural comparison of R-loop between PDB 9H4J (this study) vs 6O0Z, indicating PAM distal region increased helical pitch. **f–i**, Structural comparisons of Cas9 complexes (top) across different DNA substrates (bottom).

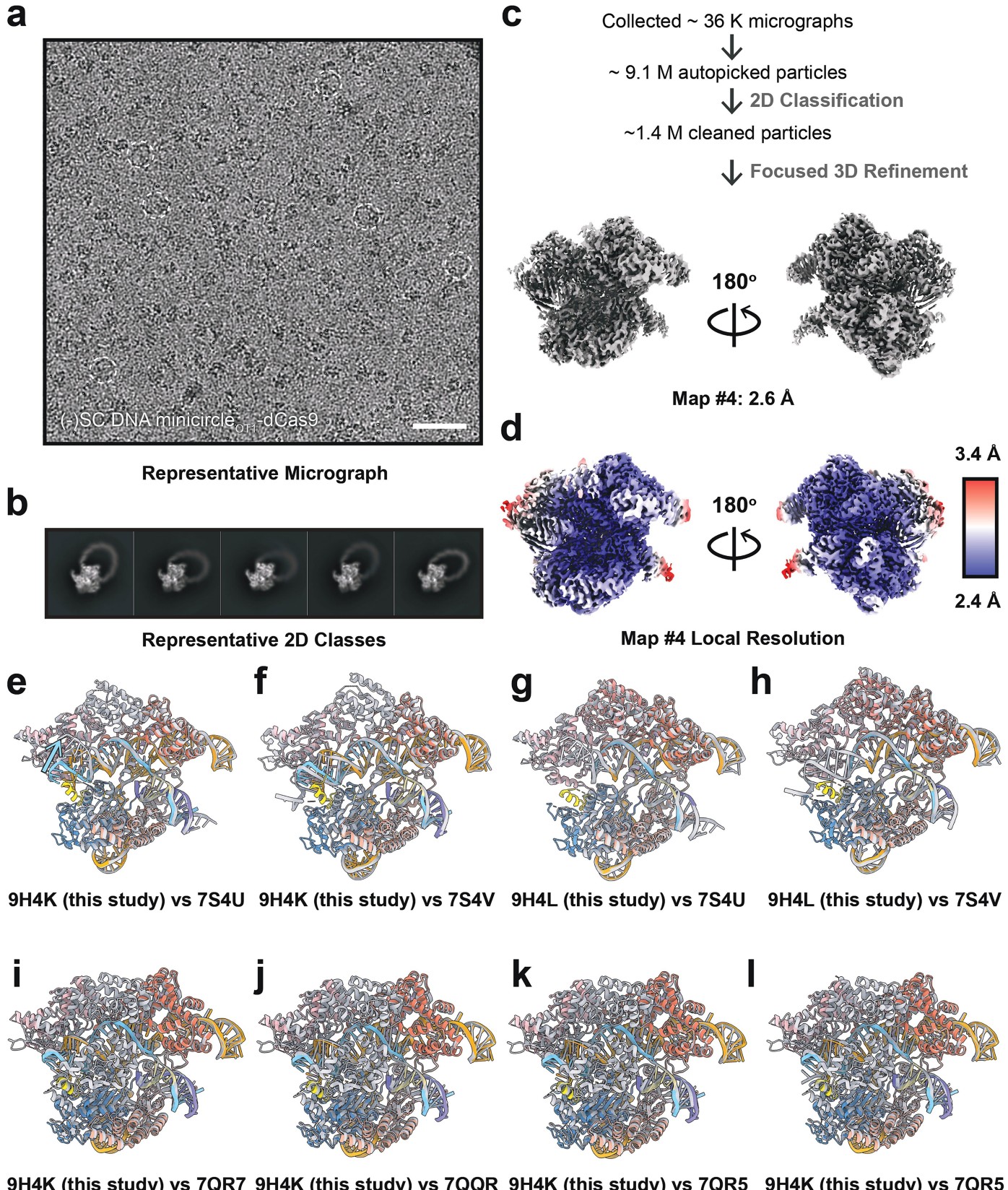

**a**
(−)SC DNA minicircle OT1 -dCas9

**Representative Micrograph**

**b**

**Representative 2D Classes**

**c**
Collected ~ 36 K micrographs
↓
~ 9.1 M autopicked particles
↓ **2D Classification**
~1.4 M cleaned particles
↓ **Focused 3D Refinement**

180°

**Map #4: 2.6 Å**

**d**

3.4 Å

2.4 Å

180°

**Map #4 Local Resolution**

**e** 9H4K (this study) vs 7S4U

**f** 9H4K (this study) vs 7S4V

**g** 9H4L (this study) vs 7S4U

**h** 9H4L (this study) vs 7S4V

**i** 9H4K (this study) vs 7QR7

**j** 9H4K (this study) vs 7QQR

**k** 9H4K (this study) vs 7QR5

**l** 9H4K (this study) vs 7QR5

**Extended Data Fig. 6 | CryoEM data processing pipeline of dCas9 complex with OT1 (−)SC DNA minicircle. a,** Representative micrograph of dCas9 complex with OT1 (−)SC DNA minicircle. **b,** Representative 2D classes, showing 'diamond ring' structure. **c,** Workflow in RELION, from micrograph collection, through 2D classification and final high-resolution 3D reconstruction. **d,** Local resolution of Map #4. **e–f,** Structural comparisons of dCas9 complex with OT1 (−)SC DNA minicircle versus 7S4U and 7S4V. **g–j,** Structural comparisons of dCas9 complex with OT2 (−)SC DNA minicircle versus 7S4U and 7S4V. **i–l,** Structural comparisons of dCas9 complex with OT1 (−)SC DNA minicircle versus structures containing seed-region purine clashes.

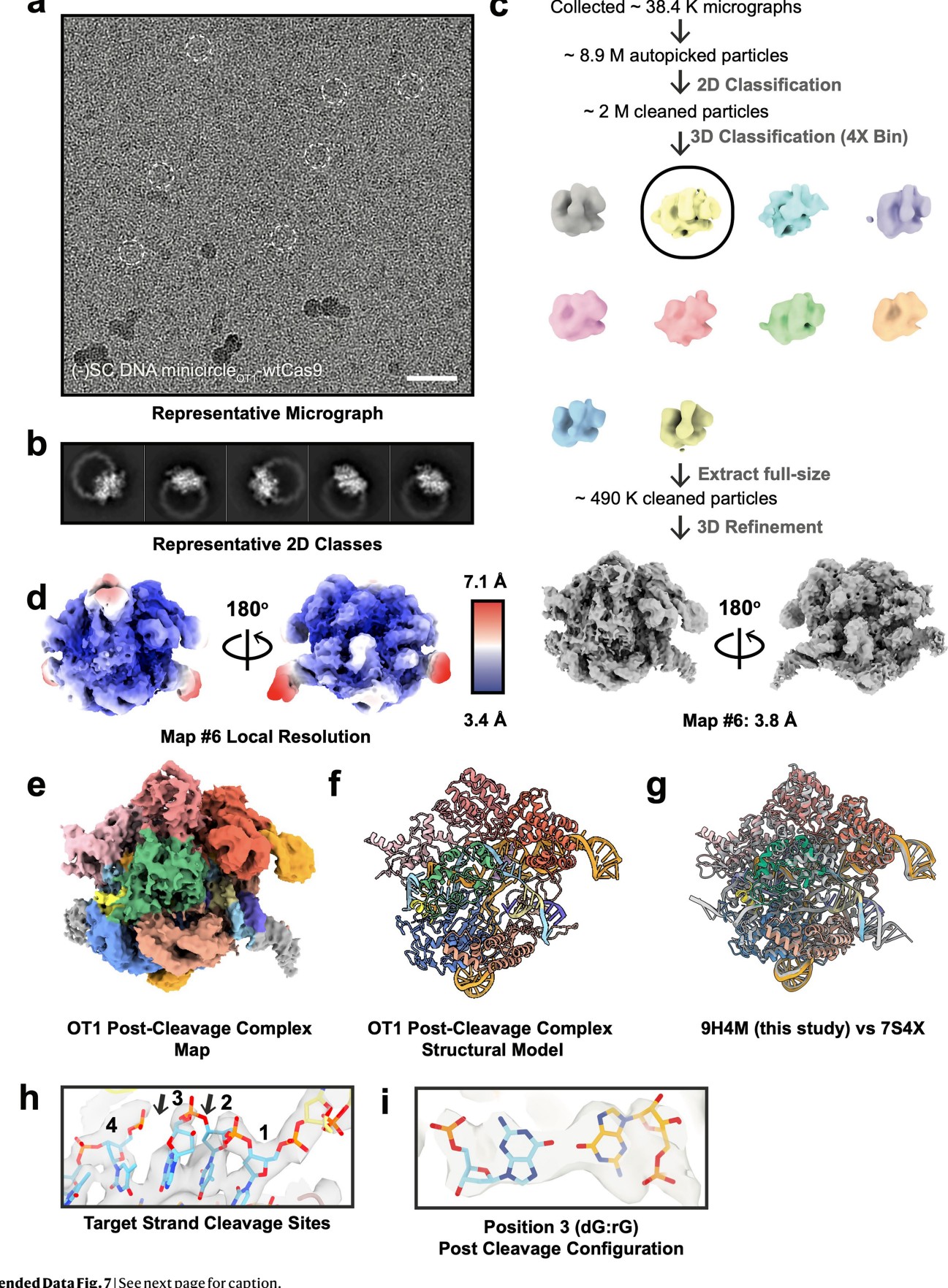

**a** Representative Micrograph

(-)SC DNA minicircle$_{OT1}$-wtCas9

**b** Representative 2D Classes

**c** Collected ~ 38.4 K micrographs

~ 8.9 M autopicked particles

**2D Classification**

~ 2 M cleaned particles

**3D Classification (4X Bin)**

**Extract full-size**

~ 490 K cleaned particles

**3D Refinement**

**d** Map #6 Local Resolution

7.1 Å

3.4 Å

180°

Map #6: 3.8 Å

180°

**e** OT1 Post-Cleavage Complex Map

**f** OT1 Post-Cleavage Complex Structural Model

**g** 9H4M (this study) vs 7S4X

**h** Target Strand Cleavage Sites

4  ↓3  ↓2  1

**i** Position 3 (dG:rG) Post Cleavage Configuration

**Extended Data Fig. 7** | See next page for caption.

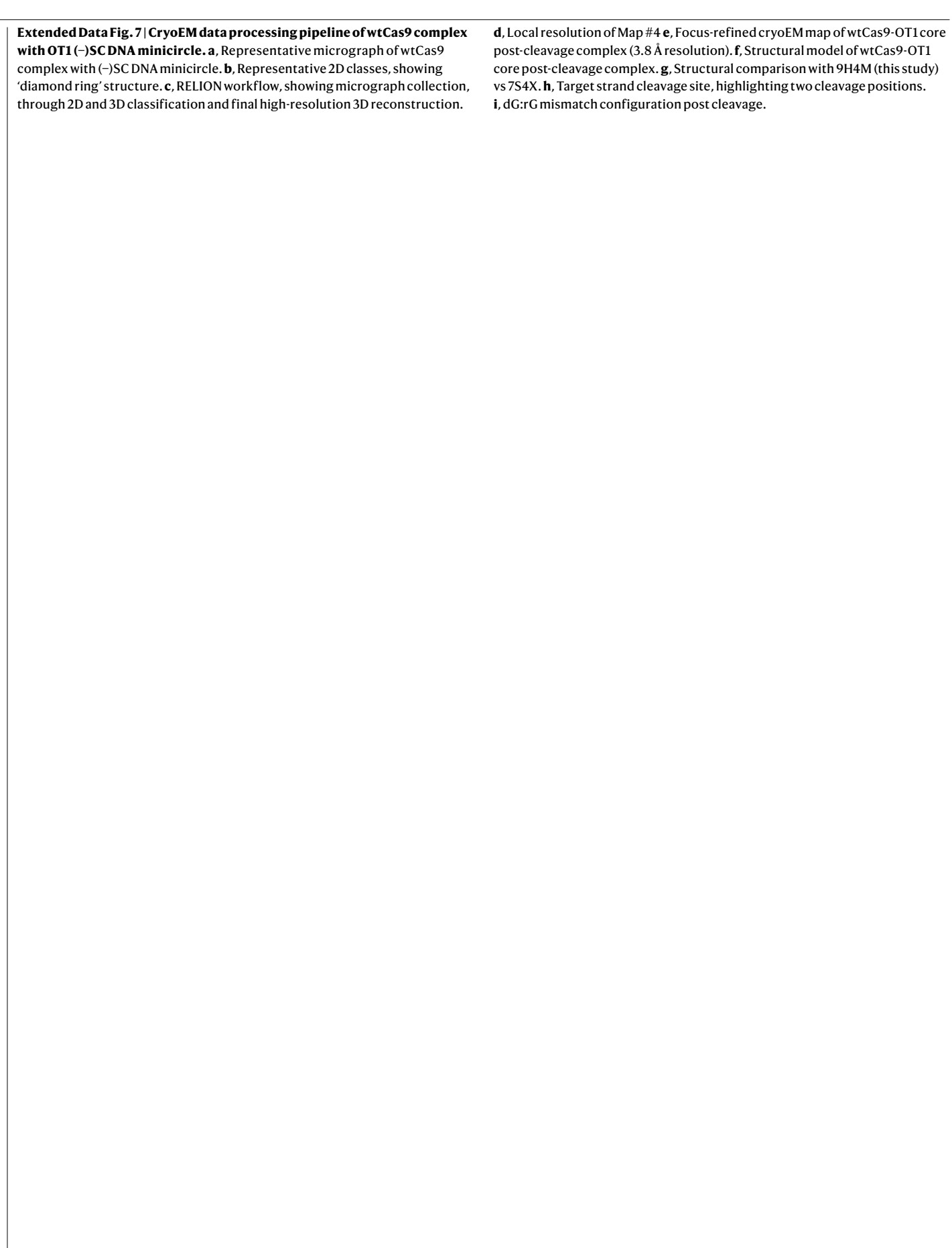

**Extended Data Fig. 7 | CryoEM data processing pipeline of wtCas9 complex with OT1 (−)SC DNA minicircle. a**, Representative micrograph of wtCas9 complex with (−)SC DNA minicircle. **b**, Representative 2D classes, showing 'diamond ring' structure. **c**, RELION workflow, showing micrograph collection, through 2D and 3D classification and final high-resolution 3D reconstruction. **d**, Local resolution of Map #4 **e**, Focus-refined cryoEM map of wtCas9-OT1 core post-cleavage complex (3.8 Å resolution). **f**, Structural model of wtCas9-OT1 core post-cleavage complex. **g**, Structural comparison with 9H4M (this study) vs 7S4X. **h**, Target strand cleavage site, highlighting two cleavage positions. **i**, dG:rG mismatch configuration post cleavage.

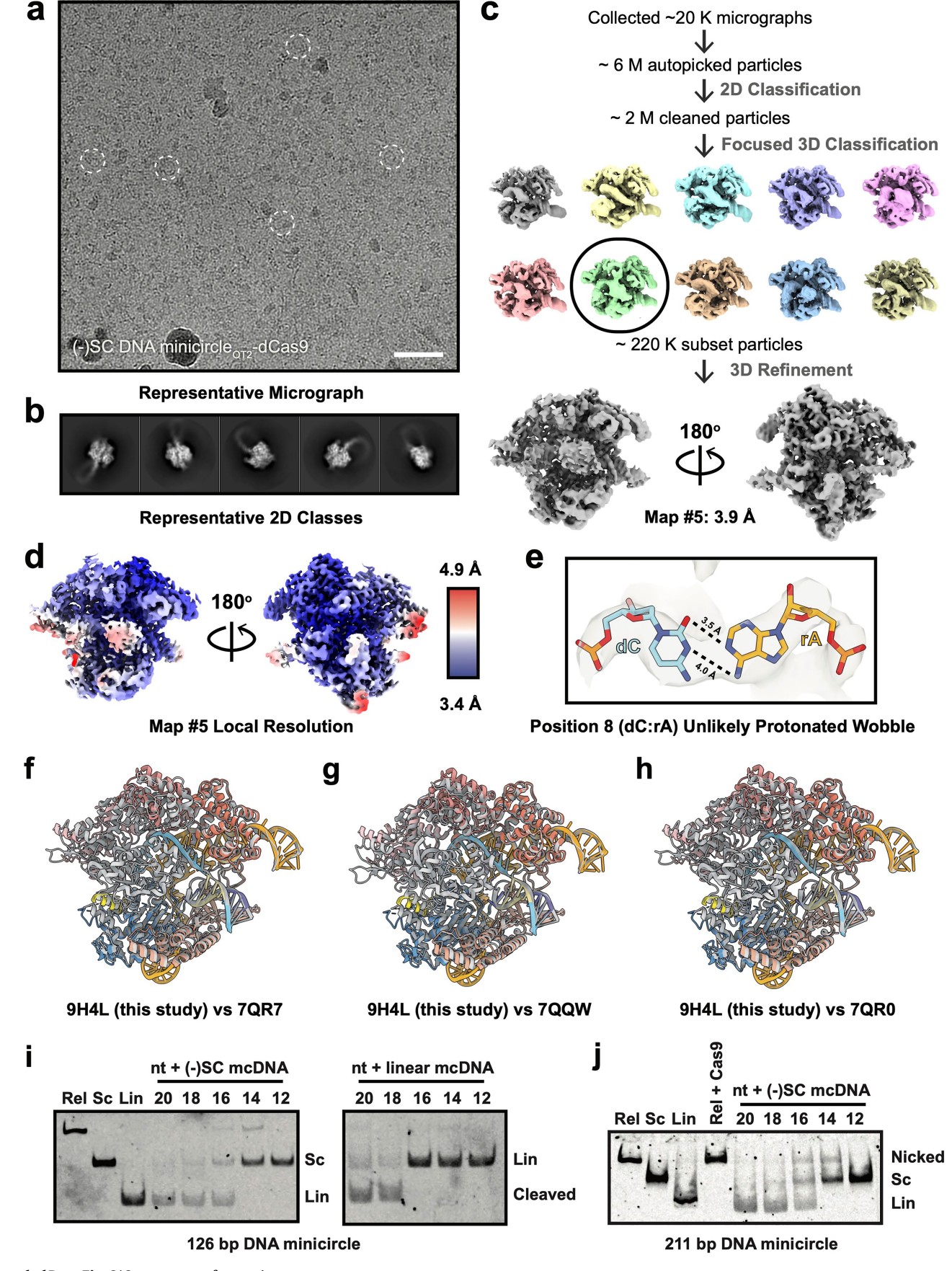

**a** Representative Micrograph

(-)SC DNA minicircle_OT2-dCas9

**b** Representative 2D Classes

**c** Collected ~20 K micrographs

~ 6 M autopicked particles
↓ 2D Classification
~ 2 M cleaned particles
↓ Focused 3D Classification

~ 220 K subset particles
↓ 3D Refinement

180°

Map #5: 3.9 Å

**d** 4.9 Å

180°

3.4 Å

Map #5 Local Resolution

**e** 3.5 Å

dC    4.0 Å    rA

Position 8 (dC:rA) Unlikely Protonated Wobble

**f** 9H4L (this study) vs 7QR7

**g** 9H4L (this study) vs 7QQW

**h** 9H4L (this study) vs 7QR0

**i** nt + (-)SC mcDNA          nt + linear mcDNA
Rel Sc Lin  20 18 16 14 12      20 18 16 14 12

Sc                              Lin
Lin                             Cleaved

126 bp DNA minicircle

**j** Rel + Cas9  nt + (-)SC mcDNA
Rel Sc Lin        20 18 16 14 12

Nicked
Sc
Lin

211 bp DNA minicircle

**Extended Data Fig. 8** | See next page for caption.

**Extended Data Fig. 8 | CryoEM data processing pipeline of dCas9 complex with OT2 (–)SC DNA minicircle. a**, Representative micrograph of dCas9 complex with (–)SC DNA minicircle. **b**, Representative 2D classes, showing 'diamond ring' structure. **c**, RELION workflow, showing micrograph collection, through 2D and 3D classification and final high-resolution 3D reconstruction. **d**, Local resolution of Map #5 **e**, A putative dC:rA protonated wobble geometry does not fit the electron density, supporting a tautomeric Watson-Crick like geometry (Fig. 4d). **f–h**, Structural comparisons of dCas9 complex with OT2 (–)SC DNA minicircle versus reported structures with multiple PAM distal mismatches. **i**, Representative 10% Native PAGE gel of Cas9 cleavage on (–)SC and linear 126 DNA minicircles (s = −0.167), quantification is on Fig. 4e. **j**, Representative 10% Native PAGE gel of Cas9 cleavage on (–)SC 211 DNA minicircles (s = −0.099), quantification is on Fig. 4e.

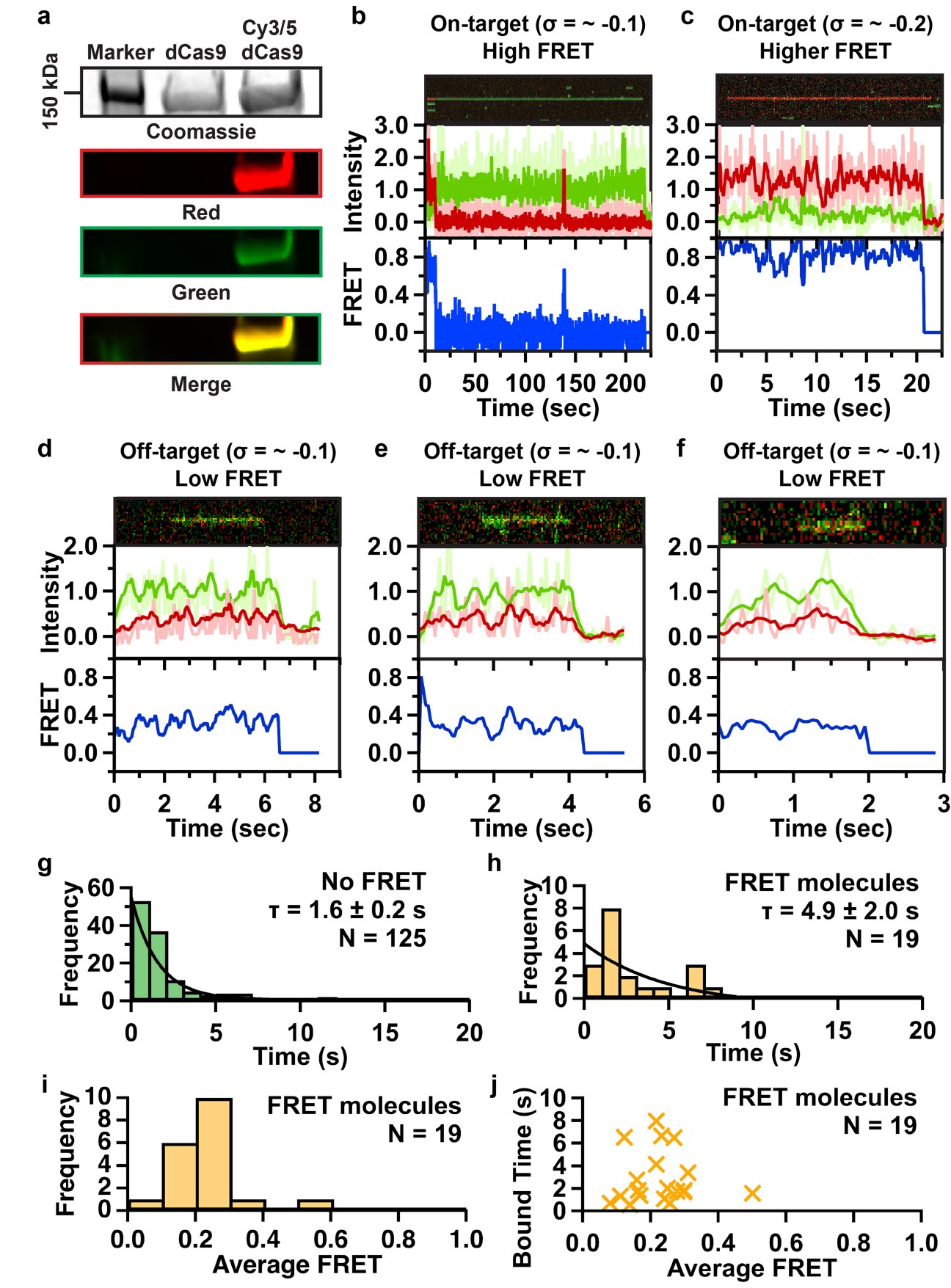

**Extended Data Fig. 9** | See next page for caption.

**Extended Data Fig. 9 | Representative FRET traces for Cas9 HNH on- and off-targets. a**, SDS-PAGE characterization of dCas9 and dCas9-cy3/cy5 substrates under Coomassie staining, and fluorescence scanning under green and red excitation. **b**–**c**, Representative kymographs, intensities and FRET profiles for on-target Cas9 bound, top = kymograph trajectory, with dotted line highlighting analysed molecule, middle = background subtracted intensities plotted across time and bottom = FRET trajectories plotted across time. **d**–**f**, Representative kymographs, intensities and FRET profiles for off-target Cas9 bound, top = kymograph trajectory, with dotted line highlighting analysed molecule, middle = background subtracted intensities plotted across time and bottom = FRET trajectories plotted across time. **g**, Dwell-time analysis of no FRET molecules (Tau = mean + error of fit). **h**, Dwell-time analysis of FRET molecules (Tau = mean + error of fit) (p-value < 0.0001 between Extended Data Fig. 9g and h) **i**, Average FRET Histogram of FRET molecules. **j**, Average FRET vs Bound Time plot of FRET molecules.

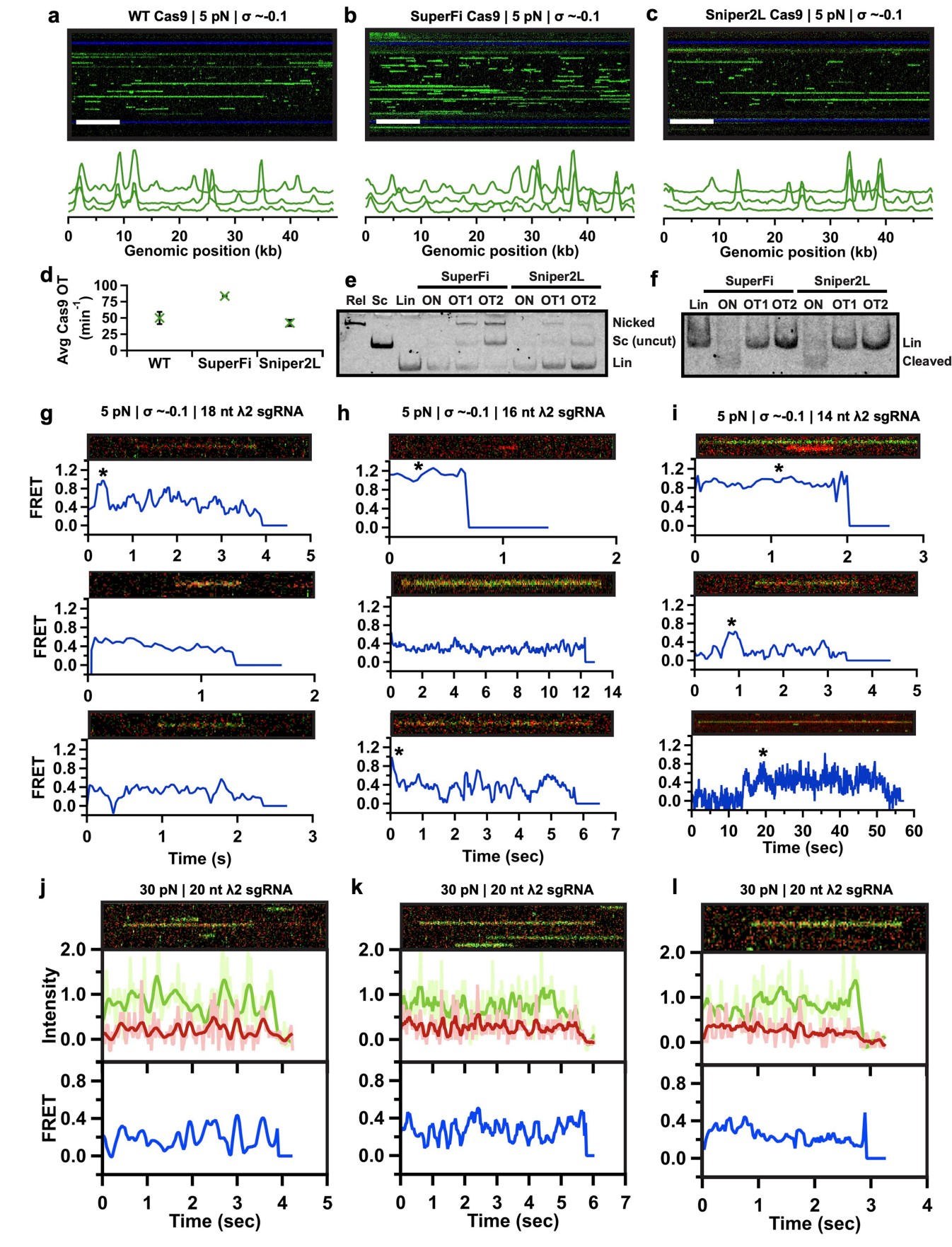

**Extended Data Fig. 10** | See next page for caption.

**Extended Data Fig. 10 | Off-target activity across high-fidelity Cas9 variants and WT Cas9 with truncated guide RNAs. a–c**, Representative kymographs of WT, SuperFi and Sniper 2 L Cas9 off-targets on (−)SC DNA with a s - −0.1, scale bar = 10 s (top) and time binned intensity histogram (bottom) showing recurring off-target binding at various genomic positions (kb). **d**, Average off-target binding events (min$^{-1}$), error bars = SEM. **e**, Representative 10% Native PAGE gel of SuperFi and Sniper2L Cas9 cleavage on (−)SC ON, OT1 and OT2 substrates. **f**, Representative 10% Native PAGE gel of SuperFi and Sniper2L Cas9 cleavage on linear ON, OT1 and OT2 substrates. **g-i**, Representative molecule for 18, 16 and 14 nt gRNA off-targets bound to s - −0.1 l-DNA (top), with FRET trajectory (bottom), asterisks indicate high FRET state achieved. **j–l**, Representative molecule for off-targets bound to l-DNA at 30 pN (top), with FRET trajectory (bottom).

**Extended Data Table 1 | Reference to small-scale producers in the 2030 Agenda and links to UN FAO Sustainable Food and Agriculture Principles**

| | On-target (EMDB-51861) (PDB 9H4J) | OT1-dCas9 (EMDB-51862) (PDB 9H4K) | OT2-dCas9 (EMDB-51863) (PDB 9H4L) | OT1-wtCas9 (EMDB-51864) (PDB 9H4M) |
|---|---|---|---|---|
| **Data collection and processing** | | | | |
| Magnification | 81,000 | 105,000 | 81,000 | 105,000 |
| Voltage (kV) | 300 kV | 300 kV | 300 kV | 300 kV |
| Electron exposure (e–/Å$^2$) | ~40 | ~40 | ~40 | ~40 |
| Defocus range (μm) | -0.5 to -2.5 | -0.5 to -2.5 | -0.5 to -2.5 | -0.5 to -2.5 |
| Pixel size (Å) | 1.1 | 0.85 | 1.1 | 0.85 |
| Symmetry imposed | C1 | C1 | C1 | C1 |
| Initial particle images (no.) | 6,393,445 | 9,069,579 | 5,547,106 | 8,911,033 |
| Final particle images (no.) | 408,016 | 1,261,170 | 2,494,515 | 496,737 |
| Map resolution (Å) | 3.1 | 2.6 | 3.9 | 3.8 |
| FSC threshold | 0.143 | 0.143 | 0.143 | 0.143 |
| Map resolution range (Å) | 2.9 to 4.4 | 2.4 to 3.4 | 3.4 to 4.9 | 3.4 to 7.1 |
| | | | | |
| **Refinement** | | | | |
| Initial model used (PDB code) | 6O0Z, AlphaFold | 9H4J, AlphaFold | 9H4J, AlphaFold | 6O0X, AlphaFold |
| Model resolution (Å) | 3.1 | 2.6 | 3.9 | 3.8 |
| FSC threshold | 0.143 | 0.143 | 0.143 | 0.143 |
| Model resolution range (Å) | 2.9 to 4.4 | 2.4 to 3.4 | 3.4 to 4.9 | 3.4 to 7.1 |
| Map sharpening $B$ factor (Å$^2$) | -80 | -200 | -150 | -90 |
| Model composition | | | | |
| Non-hydrogen atoms | 14825 | 11462 | 11937 | 13606 |
| Protein residues | 1334 | 1028 | 1164 | 1321 |
| Ligands | 0 | 0 | 0 | 0 |
| $B$ factors (Å$^2$) | | | | |
| Protein | 138.08 | 90.81 | 133.32 | 228.99 |
| Ligand | 154.95 | 98.10 | 119.52 | 115.24 |
| R.m.s. deviations | | | | |
| Bond lengths (Å) | 0.012 | 0.004 | 0.003 | 0.003 |
| Bond angles (°) | 0.881 | 0.554 | 0.663 | 0.816 |
| Validation | | | | |
| MolProbity score | 1.86 | 1.50 | 1.85 | 1.98 |
| Clashscore | 9.83 | 5.45 | 10.71 | 12.79 |
| Poor rotamers (%) | 0.08 | 1.08 | 0.77 | 0.68 |
| Ramachandran plot | | | | |
| Favored (%) | 95.03 | 96.73 | 95.65 | 94.97 |
| Allowed (%) | 4.97 | 3.27 | 4.35 | 4.95 |
| Disallowed (%) | 0.00 | 0.00 | 0.00 | 0.08 |

Cryo-EM data collection, refinement and validation statistics.

# Reporting Summary

## Statistics

For all statistical analyses, confirm that the following items are present in the figure legend, table legend, main text, or Methods section.

| n/a | Confirmed | |
|---|---|---|
| ☐ | ☒ | The exact sample size (*n*) for each experimental group/condition, given as a discrete number and unit of measurement |
| ☐ | ☒ | A statement on whether measurements were taken from distinct samples or whether the same sample was measured repeatedly |
| ☒ | ☐ | The statistical test(s) used AND whether they are one- or two-sided *Only common tests should be described solely by name; describe more complex techniques in the Methods section.* |
| ☒ | ☐ | A description of all covariates tested |
| ☐ | ☒ | A description of any assumptions or corrections, such as tests of normality and adjustment for multiple comparisons |
| ☐ | ☒ | A full description of the statistical parameters including central tendency (e.g. means) or other basic estimates (e.g. regression coefficient) AND variation (e.g. standard deviation) or associated estimates of uncertainty (e.g. confidence intervals) |
| ☒ | ☐ | For null hypothesis testing, the test statistic (e.g. *F*, *t*, *r*) with confidence intervals, effect sizes, degrees of freedom and *P* value noted *Give P values as exact values whenever suitable.* |
| ☒ | ☐ | For Bayesian analysis, information on the choice of priors and Markov chain Monte Carlo settings |
| ☒ | ☐ | For hierarchical and complex designs, identification of the appropriate level for tests and full reporting of outcomes |
| ☒ | ☐ | Estimates of effect sizes (e.g. Cohen's *d*, Pearson's *r*), indicating how they were calculated |

*Our web collection on statistics for biologists contains articles on many of the points above.*

## Software and code

Policy information about availability of computer code

| | |
|---|---|
| Data collection | Single-molecule traces of donor and acceptor intensities were acquired using Fiji (ImageJ), tracked intensities were background subtracted and plotted using Igor Pro 8, where FRET was calculated as described in previous studies. |
| Data analysis | FRET calculation, analysis and extraction was carried out manually. Fiji was used to isolate fluorescent binding and their fluorescent intensities. Igor Pro 8 was used to perform background subtraction, FRET calculations and plotting Figures for this manuscript. |

For manuscripts utilizing custom algorithms or software that are central to the research but not yet described in published literature, software must be made available to editors and reviewers. We strongly encourage code deposition in a community repository (e.g. GitHub). See the Nature Portfolio guidelines for submitting code & software for further information.

## Data

Policy information about availability of data

All manuscripts must include a data availability statement. This statement should provide the following information, where applicable:
- Accession codes, unique identifiers, or web links for publicly available datasets
- A description of any restrictions on data availability
- For clinical datasets or third party data, please ensure that the statement adheres to our policy

The datasets generated during and/or analysed during the current study will be available from the corresponding author on reasonable request.

# Research involving human participants, their data, or biological material

Policy information about studies with [human participants or human data](). See also policy information about [sex, gender (identity/presentation), and sexual orientation]() and [race, ethnicity and racism]().

| | |
|---|---|
| Reporting on sex and gender | N/A |
| Reporting on race, ethnicity, or other socially relevant groupings | N/A |
| Population characteristics | N/A |
| Recruitment | N/A |
| Ethics oversight | N/A |

Note that full information on the approval of the study protocol must also be provided in the manuscript.

# Field-specific reporting

Please select the one below that is the best fit for your research. If you are not sure, read the appropriate sections before making your selection.

☒ Life sciences  ☐ Behavioural & social sciences  ☐ Ecological, evolutionary & environmental sciences

For a reference copy of the document with all sections, see [nature.com/documents/nr-reporting-summary-flat.pdf](http://nature.com/documents/nr-reporting-summary-flat.pdf)

# Life sciences study design

All studies must disclose on these points even when the disclosure is negative.

| | |
|---|---|
| Sample size | No sample size calculation was performed. All observations were observed on sufficient numbers of individual molecules (>19). This sample size was taken from multiple different technical repeats of each condition tested.<br>In our study, we capture individual DNA molecules and probe binding and FRET in real time. |
| Data exclusions | 1) Single molecule traces that contained only acceptor signal were omitted because high background noise could bias trajectory outlining and calculated FRET efficiencies. |
| Replication | On a typical experimental day, we capture around 12 DNA molecules with roughly 12 off-targets bound. Several of these exhibit FRET characteristics. Each experiment was replicated in at least 3 independent sessions. FRET molecules exhibited similar characteristics, either toggling between low and high FRET or in a low to intermediate FRET state, results are easily reproducible. |
| Randomization | Randomization was not relevant in this study. We analyzed the fluorescence on each, individual molecule detected bound to DNA. |
| Blinding | Blinding was not relevant to this study. Experiments were performed on single molecules of fluorescently labeled protein bound to DNA. |

# Reporting for specific materials, systems and methods

We require information from authors about some types of materials, experimental systems and methods used in many studies. Here, indicate whether each material, system or method listed is relevant to your study. If you are not sure if a list item applies to your research, read the appropriate section before selecting a response.

## Materials & experimental systems

| n/a | Involved in the study |
|---|---|
| ☒ ☐ | Antibodies |
| ☒ ☐ | Eukaryotic cell lines |
| ☒ ☐ | Palaeontology and archaeology |
| ☒ ☐ | Animals and other organisms |
| ☒ ☐ | Clinical data |
| ☒ ☐ | Dual use research of concern |
| ☒ ☐ | Plants |

## Methods

| n/a | Involved in the study |
|---|---|
| ☒ ☐ | ChIP-seq |
| ☒ ☐ | Flow cytometry |
| ☒ ☐ | MRI-based neuroimaging |

## Plants

Seed stocks

N/A

Novel plant genotypes

N/A

Authentication

N/A

