## [Peer Review file · Nature]

Structural basis of supercoiling-induced CRISPR-Cas9 off-target activity

Corresponding Author: Professor David Rueda

Version 1:

Reviewer comments:

Referee #1

(Remarks to the Author)

In this manuscript, Smith et al. investigated the targeting details by Cas9 off-target on negative supercoiling ((-)SC) DNA. They resolved the structures of Cas9 in complex with both on and off-target (-)SC substrates and found that the on-target (-)SC substrate induces dCas9 to adopt a pre-catalytic state. Moreover, the off-target structures revealed that (-)SC substrate facilitates R-loop formation, suggesting (-)SC substrate promotes conformation activation. Furthermore, the authors employed single-molecule optical tweezers and FRET to demonstrate that negative supercoiling reduces the potential energy barriers for Cas9 to unwind the DNA substrate. By integrating multiple experimental techniques, the authors comprehensively explored how (-)SC substrates contribute to Cas9 off-target activity and reported a new pairing strategy between sgRNA and off-target substrates. While this study might benefit the gene-editing field, several concerns need to be addressed prior to further editorial processing.

Major comments:

1. In the second paragraph of the section “dCas9 resolves supercoiling-induced structural defects in (-)SC DNA minicircles”, the authors claim that Cas9 cleaves the on-target (-)SC substrate more rapidly than the linear substrate, based on a comparison with data from Reference 16. However, Reference 16 does not include cleavage efficiency data for the on-target site (EMX1-1) in a linear form. It is recommended that the authors perform cleavage experiments on a linear DNA substrate with the same sequence and directly compare its cleavage efficiency to that of the (-)SC substrate. Additionally, off-target cleavage should be assessed on both (-)SC and linear DNA substrates.
2. The OT2 construct contains four mismatches primarily located at the PAM-distal end, whereas OT1 harbors more mismatches within the seed region. Generally, seed region recognition plays a more crucial role in R-loop formation and subsequent cleavage than PAM-distal recognition. Could the authors explain why, in Ext. Data Fig. 1b and 1c, OT1 appears to be cleaved more efficiently than OT2? Additionally, why is a nicked band observed for OT1 but not for OT2? In the structural data, why does Cas9-OT2 exhibit larger flexibility than Cas9-OT1?
3. Could the authors clarify how these mismatched sequences were designed? Furthermore, would it be possible to conduct additional cleavage experiments using substrates with different mismatch patterns, both in linear and (-)SC DNA contexts, and compare their cleavage activity? Finally, is the enhancement of cleavage by (-)SC substrate sequence-dependent or a more general effect?
4. Compared to linear substrates, what specific features of (-)SC DNA minicircles contribute to the ~15 Å displacement of the HNH domain in the dCas9-on-target (-)SC DNA complex? Are there structural differences in the DNA strands or guide RNA between the dCas9-on-target (-)SC DNA complex and the pre-catalytic Cas9 state (PDB 6O0Z)?
5. Does DNA circularization affect R-loop formation and HNH domain dynamics? Additionally, do the authors have any

structural insights into Cas9 bound to a relaxed dsDNA minicircle?

6. Does Cas9 deactivation (dCas9 vs. Cas9) influence the conformational state of the HNH domain? The authors should compare the dCas9-on-target (-)SC DNA complex with previously reported dCas9-on-target linear dsDNA complexes to evaluate any structural differences.

7. In the section “HNH domain dynamics on off-target (-)SC DNA”, transient off-target binding events should also be measured on linear off-target DNA substrates to provide a more complete comparison.

Minor comments:

1. In Ext. Data Fig. 12e, to clarify the off-target activity of high-fidelity Cas9 variants on (-)SC DNA, cleavage experiments should also be performed on corresponding linear dsDNA substrates.

2. It is recommended to add descriptive annotations or captions to the supplementary movie to enhance clarity and comprehension.

3. In the fourth paragraph of the Discussion section, the sentence: “Furthermore, we have previously shown that unpaired ‘bubbled’ substrates and single-stranded substrates with double-stranded PAM cleave off-target faster than double-stranded linear substrates.” is not grammatically clear. Consider rephrasing for improved readability.

4. Please add line numbers in the revised manuscript.

Referee #2

(Remarks to the Author)

Smith et al. report on cryoEM structures of Cas9 bound to negatively supercoiled on-target and off-target DNA minicircles. These structures contrast previous Cas9 structures bound to linear DNA substrates and highlight how Cas9 accommodates negatively supercoiled DNA so that off-target activity is still possible. The results focus on the different types of base-pairing that occur in the R-loop to accommodate off-target sequences. In the off-target structures, the HNH domain is unable to be resolved, likely due to flexibility. Single-molecule FRET studies corroborate this hypothesis. Two higher-fidelity Cas9 proteins are also shown to bind the negatively super-coiled off-target sequences and maintain activity. These results highlight how future Cas9 engineering needs to consider DNA topology to reduce off-target activity.

This is a solid study, with the main focus of the paper being the cryoEM structures of Cas9 bound to on- and off-target substrates. These results will be of interest to those studying CRISPR-Cas systems, but also hold interest for those working with any DNA-modifying proteins, to broadly consider how native DNA topology in the cell may affect activity. The data are well-represented, with excellent figures on the cryoEM data processing pipeline for each structure and with proper statistical tests; however, error bars in Figs 1c and 4e need to be defined. Methods are well-described in sufficient detail. AFM and biochemistry studies complement the structural data showing overall changes in features of the DNA substrate and changes in cleavage activity based on nucleotide sequence. The movies in the Extended Data are an excellent addition to showcase key structural features. The conclusions drawn based on the work shown and previous studies are appropriate. It would add to the discussion if figures (in main text or Extended Data) showing the structural differences between previously published structures bound to linear off-target substrates compared to structures from this study of negatively supercoiled off-target DNA were shown to illustrate what is discussed in the second-last paragraph, along with further discussion to better highlight new features in the structures reported here compared to existing data. Overall, this was an interesting insight into the effects of DNA topology on Cas9 activity from atomic resolution structures.

Referee #3

(Remarks to the Author)

In this manuscript, Smith et al. assess the effects of negative supercoiling on R-loop formation of Cas9 with on-target and off-target DNAs. They use a multi-disciplinary approach to gain structural and functional importance of negative supercoils using atomic force microscopy (AFM), cryo-electron microscopy (cryo-EM), single-molecule FRET (smFRET), and biochemical assays. The results from all these techniques provide complementary results supporting the fact that negative superhelicity of the DNA substrate enhances DNA targeting by Cas9, and can also promote off-target effects due to the easiness in R-loop formation in negative supercoiled DNA. The novelty of the work is the determination of different cryo-EM structures of Cas9 bound to a minicircle DNA. While there is an abundance of structures of Cas9 bound to different types of linear DNA, the new structures provide a different perspective on structural activation of Cas9 that is related to the effects of DNA topology. The AFM results also show how binding of Cas9 to DNA and R-loop formation causes relaxation of the DNA minicircle, and how PAM-distal mismatches prevents DNA relaxation compared to having a complete complementarity over the 20 nt-long guide region of the guide-RNA. These results support previous observation of how PAM-distal mismatches are tolerated better by Cas9 compared to PAM-proximal mismatches.

While the cryo-EM structures of Cas9 bound to minicircle is new, the novel results coming from this work are not clearly defined. Several previous studies have shown that supercoils affect Cas9's on-target and off-target effects, including work from the same lab and from others. The cryo-EM structures also show very similar organization of the protein-nucleic acid complex compared to existing structures. Non-canonical base pairing to accommodate DNA mismatches have been described previously in other publications. Similarly, smFRET studies on Cas9 target search mechanisms and cleavage as

well as flexibility of HNH, in relation to RNA-DNA base pairing length were also previously reported. In-depth comparison of the new structures with the existing ones at different stages of conformational cascade is missing and because of this, structural features that are truly contributed by superhelicity is difficult to discern. Authors should provide a thorough analysis of the previous literature and identify new findings from this study and how superhelicity uniquely contributes to Cas9 function compared to the regular cascade of conformational activation that are needed to reach the pre-cleavage state. More details are below.

The authors state that the “structural plasticity in the PAM distal region of the protospacer are topology dependent”. Previous work using linear dsDNA templates (Jinek 2012) has also shown that when there are mismatches between 15-20 nt downstream of the PAM and the guide, there is efficient cleavage of the substrate. So, clarification is needed on how topology further augments the ability of Cas9 to cleave DNA with complete absence of complementarity beyond 15th nucleotide from the PAM.

The experiments are done using a 126-bp long minicircle DNA. In this case, binding of one Cas9 in the circle is able to relax the circle through the process of strand exchange during the R-loop formation. Binding to an off-target site with PAM-distal mismatches prevents the relaxation. This leads to several interesting questions: (i) will the effect of Cas9 on superhelicity be same when an active Cas9 that can cleave the DNA binds to a (-)SC minicircle (several off-target sites are cut by Cas9)?; (ii) will there be an effect of the length of the minicircle, which can create different levels of topological stress, on Cas9's function? (iii) Will the non-requirement of the positions beyond base pairs 12-14 change based on length of the minicircle and the ability of Cas9 to cleave DNA? Cas9 can sometimes nick off-target site, though not linearize, and this may also contribute differently to the superhelicity of the DNA. Articulation of these facts will elevate the mechanistic aspects derived from this work.

The conclusion that (-)SC causes HNH to swing closer to the active state may need further validation. The present analysis is based on PDB: 6O0Z. The authors then mention that in PDB 4O08, a similar positioning can be seen as was observed in their structure and that partial stranded DNA substrate used in PDB 4O08 may be similar to (-)SC DNA. This raises the question if the HNH placement is due to topological changes of the substrate or due to the fact that the DNA is single-stranded near the cleavage position. Both references 15 and 16 have a series of structures reported. Similarly, there are several other Cas9 structures that represent different conformational states of the complex. A systematic analysis of a set of selected structures that has used different types of DNA substrates (e.g., dsDNA, partial dsDNA, different lengths of DNA, presence/absence of divalent metal, active vs. dead Cas9) may provide details on how the positioning of HNH depends on different features of the DNA substrate.

Some discussion on how the superhelicity affects Cas9's function would be beneficial. For example, is the ssDNA region in the (-)SC DNA helping in loading of Cas9 and target search or helping in strand separation for R-loop, or both? The FRET experiment may provide more insights here based on the initial loading position vs. the PAM site with respect to different binding events that were observed. The minicircle DNA sequence seem to have a high A-T rich content. How may this affect the superhelicity of the minicircle and was the design strategic to create a minicircle that can be negatively supercoiled by gyrase?

Several types of mismatch accommodation with modified base pairing are reported in the structures bound to OT1. Of these, the G:G mismatch at position 3 has an opposite orientation of the base compared to what was observed in ref 16. Is this a random chance or it has any specific relation to (-)SC? Ref 16 shows that mismatch at position 13 has dG in anti conformation, while a mismatch at position 11 has dA in syn conformation. This shows that either the RNA or the DNA base can take an anti or syn conformation in different mismatches. A deeper analysis of the different mismatches, its position in relation to PAM in both refs 15 and 16 may provide the diversity of such non-canonical base pairing and if this is a general phenomenon or if some of the base pairing is specifically facilitated by the (-)SC of the substrate. An analysis of the flexibility of REC2 and HNH in these structures will also provide more details on how the allosteric regulation of REC2 and HNH domains are different when bound to (-)SC off-target DNA compared to a linear DNA.

OT1 has 5 mismatches covering positions 3, 8, 9, 15, 20 (including seed and PAM distal), while OT2 has only 4 mismatches covering positions 8, 15, 16, 20 (lesser number of seed mismatches). The PAM-distal cryo-EM density in OT2 is also poor indicating flexibility in this region. The authors state that having three mismatches at the PAM-distal end maybe causing this flexibility. It can also be interpreted that having three PAM-distal mismatches with the RNA in OT2 may make the dsDNA more stable at this end, compared to having only two mismatches in OT1. It is also stated that “The absence of discernable density in the nine PAM-distal bp of the R-loop raises the interesting possibility that the increased flexibility of the (-)SC DNA bases decreases the requirement of a complete R-loop formation for cleavage activity.”. Related to this, ref 16 has used only a short NTS strand and a full-length TS strand for crystallization, and this partial dsDNA may give a more flexible PAM-distal end compared to having a (-)SC. How different are the structures of OT2 compared to some of the off-targets from ref 16? Ref 16 also states that there is unpairing at the PAM-distal end DNA when there are mismatches at this region. In short, a thorough analysis of the diverse Cas9 structures with mismatches will help in delineating the differences that are arising exclusively from the superhelicity of the DNA. The AFM data on binding of dCas9 to different minicircles may need further explanation. In the main figures, only one type of minicircle is shown, which maybe that of the on-target DNA (the legend is not clear on this). In ED Fig. 3, there is data for on-target, OT1 and OT2, but the legend or the main text does not explain the conclusion from the data. Is there any difference in the amount of DNA relaxation between OT1 and OT2 sites? There are not many binding events seen in panel c for OT2.

In ED Fig. 8f, the amount of nicking for sgRNA with only 14-nt guide is very low. The nicking is proposed to be related to the 14-nt bubble causing off-target as measured by CIRCLE-seq in ref 24. Nicked DNA can be repaired by human cells, and

hence only nicking potential may not directly relate to off-target effects. The statement "In recent linear off-target structures 15,16, PAM-distal mismatches were clearly resolved, supporting the idea that Cas9 cleavage activity with short guides (~14 nt) is specific to (-)SC DNA substrates.", is not clear. How is resolution of the base pairs related to DNA cleavage efficiency?

How are the docking positions of the HNH different than what was previously reported using truncated RNAs (DOI: 10.1126/sciadv.aao002)? Some direct comparisons of the previous results are needed to show the effects coming only from (-)SC. A 17-nt long checkpoint was determined in this previous work, which showed that only after reaching this position, HNH can achieve the activated state. Will the (-) SC reduce the number of base pairs needed for this checkpoint? Does this number differ based on on-target and off-target DNA substrates in the case of (-) SC and also based on the length of the minicircle?

The statement, "These results show that Cas9 high-fidelity variants derived from structures in complex with linear DNA substrates are unlikely to maintaining their high fidelity in the context of (-)SC DNA, thereby justifying the use of (-)SC minicircles to further improve Cas9 fidelity.", may need some modification. Several of the high-fidelity variants were developed by direct evolution which does not specifically modify positions based on structural interaction. In addition, these variants are tested for their ability to edit genes in cell lines, which represent native genomic organization with (-) SC DNA. So, the directly evolved high-fidelity variants are being selected based on real interactions with cellular DNA, not based on information from a molecular structure.

Showing an overlay of the new structures with some from ref 16 may help to assess if the movement of the domains are due to only topology or the different structures are being captured at different intermediate steps towards reaching the precatalytic state.

Minor points

Fig. 3: Show positions for HNH and REC3, maybe using dashed circles?

Please specify if Fig. 4a is OT1 with or without Mg²⁺. In Fig. 3, HNH is not visible, in 4a it is. It is better to clarify this in the legend since there are two OT1 structures being reported. Is OT2 pre- or post-cleavage? If OT2 is pre-cleavage, why are the post- and pre-cleavage stages being compared in this figure? It should both be the same state to show the difference in domain placements due to the effect of off-target at different positions of the protospacer.

Extended data shows different maps, labelled map #1, 2, 3, 4, 5 etc. for different figures. Map 3 is listed in both ED Figs 6 and 7. Are these maps from different population of particles? The numbering scheme is not clear.

In ED Fig. 11, what is the time for panel e?

For ED Fig. 12, please explain the image and corresponding graph that shows different binding events, specifying how they relate to different variants and off-target binding. This will help a non-expert to understand this data better. Did the DNA used in this experiment hold both on-target and off-target sites? How does the on-target binding compare to off-target binding in the different variants?

Referee #4

(Remarks to the Author)

Smith et al. present in their manuscript a beautiful set of structures of Cas9 bound to initially negatively supercoiled minicircles where even the minicircles are resolved. Even from an esthetic point of view it is fun to see this. As main findings they show that for fully matching target DNA the HNH domain engaged closer to the TS scissile phosphate compared to relaxed DNA. Furthermore they present two structures on off-targets containing multiple mismatches, one with a full R-loop (OT1) and one with a partial R-loop (OT2). For some of the resolved mismatches they obtain new configurations that have not been observed before. Furthermore, they use fluorescence experiments on supercoiled DNA to show that the HNH domain is highly dynamic on supercoil-induced off-targets. Overall this is a very interesting and carefully executed study which is also very well presented. I have therefore only very few technical remarks. I am convinced that the work will be highly interesting for the Cas9 community, in particular to researchers focusing on the structural aspects of Cas9.

Detailed remarks:

- How relevant are the applied supercoiling densities in this study (absolute values of ≥ 0.1) compared to supercoiling levels in eukaryotic systems where the DNA is rather relaxed (in contrast to prokaryotic systems with supercoiling densities of around -0.05)? Are there any reliable quantitative numbers for the supercoiling densities in eukaryotes in order to evaluate whether supercoiling can provide a detectable impact on off-targeting in these systems?

- Are the unique mismatch configurations on OT1 and OT2 observed in this study really supercoil-specific or would they also be observable on relaxed DNA when using the same base-pair neighbors? In other words, is the occurrence of the new mismatch configurations just caused by the so far limited number of available mismatch structures?

- The authors state that "negative supercoiling reduces the potential energy barriers to unwind the DNA substrate (dsDNA) and facilitate R-loop formation, even in the presence of multiple mismatches". In this context they should refer to previous studies on CRISPR-Cas effectors that evaluated quantitatively how mismatches are overcome as a function of the applied

supercoiling and how this can be effectively modelled/understood.

- The HNH dynamics is still somewhat confusing to me. While I understand that the HNH domain can be well resolved in structures with a fully matching target, I don't understand why it appears to be flexible for the uncleaved OT1 target despite the presence of a full R-loop. Are there any specific contacts of the HNH domain to the heteroduplex that would be not established if the heteroduplex contains mismatches?

- Ext. Data Fig. 11: D) The obtained decay time of $\tau = 0.6$ s seems to be inconsistent with plotted data and fit function. At τ the number of FRET molecules should have decreased to $1/e$ (about $(1/3)$ of the initial amplitude. Neither data nor fit show this, however. The errors for τ are in my opinion too high in D) but too low in F) given the high and low event numbers, respectively. The standard deviation of a normalized exponential distribution is τ , such that the standard error of τ should be given by τ/\sqrt{N} . This way the error in D should be about 0.06 s while in F about 1s. The authors should then test whether the difference of τ for the two different event types is still significant.

Version 2:

Reviewer comments:

Referee #1

(Remarks to the Author)

The authors have addressed all of our questions, and I have no further major concerns.

1. On p. 3, line 25, "...because (-)SC the DNA substrate..." should be revised to "...because (-)SC DNA substrate...".

2. In Response 2, the reference to "the text on Page 11, lines 25-27" appears to be incorrect.

3. In Response 5, the statement "The data show that relaxed DNA minicircles do not form stable R-loops (Extended Data Figure 1j)" should be updated to refer to "Extended Data Figure 11,m."

Referee #3

(Remarks to the Author)

The authors have made a tremendous effort in addressing my queries from the first submission. Comparison of the new structures with several previous ones truly clarifies the effects on superhelicity on on-target and off-target DNA cleavages. I also appreciate the added biochemical and binding studies that also confirmed the effect of superhelicity on DNA binding events as well as on the relaxed checkpoint needed for negatively supercoiled DNA compared to linear DNA. I do not have any major comments, just one below to check if a figure citation is correct.

Minor comments:

Organizing figures in the order of appearance in the main text will improve readability. Ext. Data Fig. 8 is cited in the main text before Ext. Data Fig. 7.

Please check lines 29-31 of p. 6 ("While the entire DNA ring is still visible, it now adopts a 'teardrop' conformation (Extended Data Fig. 8b), indicating that the DNA is not fully relaxed upon dCas9 binding, consistent with partial R-loop formation."). Is this Ext. Data Fig. 8b or Ext. Data Fig. 7b? Switching current Ext. Data Fig. 7b to Ext. Data Fig. 8b will make the flow align better with the order things are mentioned in the main text.

Referee #4

(Remarks to the Author)

The authors have well addressed my previous remarks/concerns such that I recommend publication of the manuscript.

The only point that was not correctly addressed is my previous final remark:

Ext. Data Fig. 11: D) The obtained decay time of $\tau = 0.6$ s seems to be inconsistent with plotted data and fit function. At τ the number of FRET molecules should have decreased to $1/e$ (about $(1/3)$ of the initial amplitude. Neither data nor fit show this, however. The errors for τ are in my opinion too high in D) but too low in F) given the high and low event numbers, respectively. The standard deviation of a normalized exponential distribution is τ , such that the standard error of τ should be given by τ/\sqrt{N} . This way the error in D should be about 0.06 s while in F about 1s. The authors should then test whether the difference of τ for the two different event types is still significant.

While the authors have corrected the values for τ in both plots, the error in the Ext. Data Fig. 9h does still not make sense. It is impossible to obtain an error of 0.02 s for an exponential decay with a mean of 4.87 s from just 19 measured values. Least mean square fitting tends to underestimate the errors since it uses as data errors the actual rms. I explained previously how the error could be estimated. In the light of this I doubt that the obtained p value of 0.0001 is correct. When using a t-test, the authors should consider that the two distributions have very different sample sizes and also different standard deviations. Furthermore, the p value is not provided in the caption of Ext. Data Fig. 9 as promised by the authors.

Referees' comments

Referee #1:

In this manuscript, Smith et al. investigated the targeting details by Cas9 off-target on negative supercoiling ((-)SC) DNA. They resolved the structures of Cas9 in complex with both on and off-target (-)SC substrates and found that the on-target (-)SC substrate induces dCas9 to adopt a pre-catalytic state. Moreover, the off-target structures revealed that (-)SC substrate facilitates R-loop formation, suggesting (-)SC substrate promotes conformation activation. Furthermore, the authors employed single-molecule optical tweezers and FRET to demonstrate that negative supercoiling reduces the potential energy barriers for Cas9 to unwind the DNA substrate. By integrating multiple experimental techniques, the authors comprehensively explored how (-)SC substrates contribute to Cas9 off-target activity and reported a new pairing strategy between sgRNA and off-target substrates. While this study might benefit the gene-editing field, several concerns need to be addressed prior to further editorial processing.

Major comments:

1. In the second paragraph of the section “dCas9 resolves supercoiling-induced structural defects in (-)SC DNA minicircles”, the authors claim that Cas9 cleaves the on-target (-)SC substrate more rapidly than the linear substrate, based on a comparison with data from Reference 16. However, Reference 16 does not include cleavage efficiency data for the on-target site (EMX1-1) in a linear form. It is recommended that the authors perform cleavage experiments on a linear DNA substrate with the same sequence and directly compare its cleavage efficiency to that of the (-)SC substrate. Additionally, off-target cleavage should be assessed on both (-)SC and linear DNA substrates.

We agree with the reviewer. To address this, we have performed ensemble cleavage assays on linearised on- and off-target substrates (see Extended Figure 1f-k). Cas9 cleaves the on-target DNA substrates (linear and (-)SC) rapidly. Interestingly, Cas9 does not cleave either OT1-OT4 in linear form, supporting our main hypothesis that (-)SC substrates accelerate off-target cleavage. We have now clarified this on Page 3, line 20-21, and included the new data as Extended Figure 1h-k.

2. The OT2 construct contains four mismatches primarily located at the PAM-distal end, whereas OT1 harbors more mismatches within the seed region. Generally, seed region recognition plays a more crucial role in R-loop formation and subsequent cleavage than PAM-distal recognition. Could the authors explain why, in Ext. Data Fig. 1b and 1c, OT1 appears to be cleaved more efficiently than OT2?

We thank the reviewer for raising this important point. In the context of (-)SC substrates, the seed region becomes less crucial in R-loop formation, and more mismatches are tolerated in it. We showed this in our previous manuscript (Mol Cell 2023, PMID: 37802026), which is why we chose these specific off-targets (OT1 and OT2). The idea is that (-)SC DNA substrates lower the local melting

energy barriers, thereby facilitating R-loop formation and increasing mismatch tolerance. To clarify this point, we modified the manuscript on Page 3, lines 23-27.

Additionally, why is a nicked band observed for OT1 but not for OT2?

The effect of (-)SC DNA substrates (and their sequence) in accelerating the first and second cleavage steps is not necessarily the same. In the case of OT1 (which forms a complete R-loop), the first cleavage step is faster than the second one, resulting in an observable nicked band. However, this is not the case for OT2 (which does not form a complete R-loop). We have now clarified this in the manuscript on Page 3, lines 27-30.

In the structural data, why does Cas9-OT2 exhibit larger flexibility than Cas9-OT1?

This is because OT2 has primarily PAM-distal mismatches, whereas OT1 has mismatches distributed throughout the protospacer. We have now clarified this in the text on Page 11, lines 25-27.

3. Could the authors clarify how these mismatched sequences were designed?

Both off-targets were selected from our previous CIRCLE-seq experiments (Newton *et al.* Mol Cell 2023, PMID: 37802026). OT1 was chosen for its high number (5) of mismatches, distributed mostly in the seed region (PAM-proximal). Whereas OT2 was chosen for its high number (4) of mismatches, which are distributed mostly in the PAM-distal region. We also considered the types of mismatches in both cases (e.g., Wobble pairs, pyrimidine-pyrimidine, and purine-purine clashes), as described in the text. We have now clarified these criteria in the revised manuscript (Page 3, lines 8-9).

Furthermore, would it be possible to conduct additional cleavage experiments using substrates with different mismatch patterns, both in linear and (-)SC DNA contexts, and compare their cleavage activity?

We have selected two additional off-targets (OT3 and OT4) from our previous CIRCLE-seq experiments (Newton *et al.* Mol Cell 2023, PMID: 37802026) with unique mismatch combinations: OT3 and OT4 have six and four mismatches, respectively (see Extended Data Figures 1d-e). The resulting (-)SC cleavage rate constants are $0.09 \pm 0.01 \text{ min}^{-1}$ and $0.07 \pm 0.02 \text{ min}^{-1}$, respectively, while the linearised substrates don't exhibit any cleavage (Extended Data Figure 1j-k). These new data show that the enhancement of cleavage is likely a general effect, in agreement with our previous CIRCLE-seq data. We have now included this in the revised manuscript (Page 3, lines 31-33)

Finally, is the enhancement of cleavage by (-)SC substrate sequence-dependent or a more general effect?

The original manuscript data, together with the new additional experiments with OT3 and OT4, strongly support the idea that off-target cleavage enhancement by substrate supercoiling is indeed a general effect. However, the sequence of the target can still play a role in modulating the cleavage rates (e.g. OT1 and OT2

have different first/second cleavage step kinetics). We have now clarified in the text (Page 3, lines 33-35).

4. Compared to linear substrates, what specific features of (-)SC DNA minicircles contribute to the ~15 Å displacement of the HNH domain in the dCas9-on-target (-)SC DNA complex? Are there structural differences in the DNA strands or guide RNA between the dCas9-on-target (-)SC DNA complex and the pre-catalytic Cas9 state (PDB 6O0Z)?

To address this point, we have performed a more in-depth comparison of the structural differences between our structure and the previous pre-catalytic Cas9 state (PDB 6O0Z). We had previously noted key differences in the position of the HNH domain (closer to the scissile phosphate), the REC3 domain, which appears to clamp more tightly onto the protospacer, and the REC2 domain, which moves away from the protospacer to make space for the HNH domain. Following the reviewer's suggestion, we have now also noticed that the PAM-distal region of the R-loop exhibits an extended helical pitch compared to a linear substrate. This is likely due to the increased flexibility of the (-)SC DNA substrate compared to the linear one. To clarify this, we have now revised the manuscript (Page 5, lines 26-32) and included the (Extended Data Figure 5e).

5. Does DNA circularization affect R-loop formation and HNH domain dynamics? Additionally, do the authors have any structural insights into Cas9 bound to a relaxed dsDNA minicircle?

This is an important point. Our hypothesis is that (-) supercoiling, not circularisation alone, enhances R-loop formation and promotes allosteric activation of the HNH domain. To test this, we performed electrophoretic mobility shift assays (EMSAs) with dCas9 on relaxed and (-)SC DNA minicircles. The data show that relaxed DNA minicircles do not form stable R-loops (Extended Data Figure 1j), demonstrating that (-) supercoiling, not circularisation, facilitates Cas9 R-loop formation and off-target activity. We have now clarified in the text (Page 3, lines 39-43) and added the new data as Extended Figures 1l-m.

6. Does Cas9 deactivation (dCas9 vs. Cas9) influence the conformational state of the HNH domain? The authors should compare the dCas9-on-target (-)SC DNA complex with previously reported dCas9-on-target linear dsDNA complexes to evaluate any structural differences.

There is one structure of dCas9 bound to linear DNA (Huai et al. (2017) Nat Commun. PMID: 29123204). Unfortunately, the resolution in this structure is only 5.2 Å, which prevents us from making meaningful comparisons regarding the conformational state of the HNH domain. We have now clarified in the text (Page 10, lines 14-15).

7. In the section "HNH domain dynamics on off-target (-)SC DNA", transient off-target binding events should also be measured on linear off-target DNA substrates to provide a more complete comparison.

To address this point, we have performed new optical tweezers experiments on linear DNA at high force (30 pN) with the doubly labelled dCas9 construct. The resulting kymographs (Response Figure 1 below) show similar dynamics of the HNH domain as in stretched (-)SC DNA. We have now added the new data as Extended Data Figure 10j-l and clarified this in the text (Page 8, lines 39-41).

Response Figure 1. Off-target HNH dynamics on linear force-stretched DNA. (Top) Representative Kymographs of three (a-c) off-target dCas9 molecules bound to force-stretched (30 pN) λ -DNA. (Middle) Corresponding donor (green) and acceptor (red) fluorescence intensities showing anticorrelated fluorescence. (Bottom) Calculated FRET trajectories showing HNH domain conformational dynamics.

Minor comments:

1. In Ext. Data Fig. 12e, to clarify the off-target activity of high-fidelity Cas9 variants on (-)SC DNA, cleavage experiments should also be performed on corresponding linear dsDNA substrates.

To address this point, we have performed cleavage assays on linear DNA with Sniper2L and SuperFi (Extended Data Figure 10f). Both high-fidelity variants exhibit on-target cleavage but no off-target cleavage on linear DNA. This data further supports the hypothesis that (-)SC regulates off-target cleavage. We have now clarified in the text (Page 9, lines 24-25).

2. It is recommended to add descriptive annotations or captions to the supplementary movie to enhance clarity and comprehension.

Done.

3. In the fourth paragraph of the Discussion section, the sentence: "Furthermore, we have previously shown that unpaired 'bubbled' substrates and single-stranded substrates with double-stranded PAM cleave off-target faster than double-stranded

linear substrates.” is not grammatically clear. Consider rephrasing for improved readability.

Done.

4. Please add line numbers in the revised manuscript.

Done.

Referee #2:

Smith et al. report on cryoEM structures of Cas9 bound to negatively supercoiled on-target and off-target DNA minicircles. These structures contrast previous Cas9 structures bound to linear DNA substrates and highlight how Cas9 accommodates negatively supercoiled DNA so that off-target activity is still possible. The results focus on the different types of base-pairing that occur in the R-loop to accommodate off-target sequences. In the off-target structures, the HNH domain is unable to be resolved, likely due to flexibility. Single-molecule FRET studies corroborate this hypothesis. Two higher-fidelity Cas9 proteins are also shown to bind the negatively super-coiled off-target sequences and maintain activity. These results highlight how future Cas9 engineering needs to consider DNA topology to reduce off-target activity.

This is a solid study, with the main focus of the paper being the cryoEM structures of Cas9 bound to on- and off-target substrates. These results will be of interest to those studying CRISPR-Cas systems, but also hold interest for those working with any DNA-modifying proteins, to broadly consider how native DNA topology in the cell may affect activity. The data are well-represented, with excellent figures on the cryoEM data processing pipeline for each structure and with proper statistical tests; however, error bars in Figs 1c and 4e need to be defined.

We thank the reviewer for their kind comments and for bringing this to our attention. The error bars are now defined as standard error of the mean (SEM) (Figure 1c and 4e).

Methods are well-described in sufficient detail. AFM and biochemistry studies complement the structural data showing overall changes in features of the DNA substrate and changes in cleavage activity based on nucleotide sequence. The movies in the Extended Data are an excellent addition to showcase key structural features. The conclusions drawn based on the work shown and previous studies are appropriate. It would add to the discussion if figures (in main text or Extended Data) showing the structural differences between previously published structures bound to linear off-target substrates compared to structures from this study of negatively supercoiled off-target DNA were shown to illustrate what is discussed in the second-last paragraph, along with further discussion to better highlight new features in the structures reported here compared to existing data.

We agree with the reviewer's suggestion. To address this, we have expanded our discussion by comparing our structures to other structures.

We have performed a more in-depth comparison of the structural differences between our on-target structure (PDB 9H4J) and the previous pre-catalytic Cas9 state (6O0Z). We had previously noted key differences in the position of the HNH domain (closer to the scissile phosphate), the REC3 domain, which appears to clamp more tightly onto the protospacer, and the REC2 domain, which moves away from the protospacer to make space for the HNH domain. Following the reviewers' suggestions, we have now also noticed that the PAM-distal region of the R-loop exhibits an extended helical pitch compared to a linear substrate. This is likely due to the increased flexibility of the (-)SC DNA substrate compared to the

linear one. To clarify this, we have now revised the manuscript (Page 5, lines 26-32) and included the (Extended Data Figure 5e).

We also include a comparison of the OT1 structure (9H4K) to the pre-catalytic structure (7S4U, Bravo et al. (2022), PMID: 29123204). The Cas9 architecture remains overall unchanged, but the REC2 domain is visible in their structure. The key differences between the two structures are located in the PAM distal region of the R-loop. In 7S4U, the PAM-distal DNA is distorted upwards towards the REC3 domain with a local RMSD of ~ 14 Å across the TS backbone (Extended Data Figure 6e), whereas in our OT1 structure, the R-loop is already completely formed. This is consistent with the proposed idea that the R-loop forms faster on (–)SC substrates compared to linear DNA. We have included this on Page 6, lines 36-43.

We also include a comparison between our OT1 (9H4K) structure and structures containing seed-region purine clashes (A:A, G:A and G:G in Pacesa et al (2022) Cell, PMID: 36306733). While the overall Cas9 architecture remains nearly identical, in our structure, the REC3 domain appears to be more outward by ~ 5.4 Å than in any of the four seed-region purine clash structures (PDBs 7QR7, 7QQR, 7QQQ, 7QQX, Extended Data Figure 6i-l). Although the REC3 domain has been implicated in R-loop proofreading (Skeens et al. (2024) Sci Adv., PMID: 38446895), Cas9 cleaves all these linear off-targets at least 10-fold slower than OT1 under (–)SC conditions. Therefore, these results are consistent with the idea that mismatches on (–)SC substrates are more tolerated than on linear DNA.

We also compare the OT2 structure (9H4L) with three structures containing 3 to 5 PAM-distal mismatches in the same study (Pacesa et al., PDBs 7QR7, 7QQW, 7QR0). Again, while the overall Cas9 structure remains similar, the REC3 domain in our structure appears to be ~ 5.4 Å more outward than in any of the three PAM-distal mismatches structures (Extended Data Figure 7f-h). However, Cas9 cleaves these linear off-targets at least 10-fold slower than OT2 under (–)SC conditions, consistent with the idea that PAM-distal mismatches on (–)SC substrates are more tolerated than on linear DNA.

Finally, we added a comparison of our OT1 product structure (9H4M) with the existing product structure with a linear substrate (7S4X, Bravo et al. (2022) Nature, PMID: 35236982). The overall Cas9 architecture remains the same between both structures, consistent that Cas9 holds onto the DNA substrate post-cleavage (Extended Data Figure 8g).

Overall, this was an interesting insight into the effects of DNA topology on Cas9 activity from atomic resolution structures.

We thank the reviewer again for their kind comments.

Referee #3:

In this manuscript, Smith et al. assess the effects of negative supercoiling on R-loop formation of Cas9 with on-target and off-target DNAs. They use a multi-disciplinary approach to gain structural and functional importance of negative supercoils using atomic force microscopy (AFM), cryo-electron microscopy (cryo-EM), single-molecule FRET (smFRET), and biochemical assays. The results from all these techniques provide complementary results supporting the fact that negative superhelicity of the DNA substrate enhances DNA targeting by Cas9, and can also promote off-target effects due to the easiness in R-loop formation in negative supercoiled DNA. The novelty of the work is the determination of different cryo-EM structures of Cas9 bound to a minicircle DNA. While there is an abundance of structures of Cas9 bound to different types of linear DNA, the new structures provide a different perspective on structural activation of Cas9 that is related to the effects of DNA topology. The AFM results also show how binding of Cas9 to DNA and R-loop formation causes relaxation of the DNA minicircle, and how PAM-distal mismatches prevents DNA relaxation compared to having a complete complementarity over the 20 nt-long guide region of the guide-RNA. These results support previous observation of how PAM-distal mismatches are tolerated better by Cas9 compared to PAM-proximal mismatches.

While the cryo-EM structures of Cas9 bound to minicircle is new, the novel results coming from this work are not clearly defined. Several previous studies have shown that supercoils affect Cas9's on-target and off-target effects, including work from the same lab and from others. The cryo-EM structures also show very similar organization of the protein-nucleic acid complex compared to existing structures. Non-canonical base pairing to accommodate DNA mismatches have been described previously in other publications. Similarly, smFRET studies on Cas9 target search mechanisms and cleavage as well as flexibility of HNH, in relation to RNA-DNA base pairing length were also previously reported. In-depth comparison of the new structures with the exiting ones at different stages of conformational cascade is missing and because of this, structural features that are truly contributed by superhelicity is difficult to discern. Authors should provide a thorough analysis of the previous literature and identify new findings from this study and how superhelicity uniquely contributes to Cas9 function compared to the regular cascade of conformational activation that are needed to reach the pre-cleavage state. More details are below.

We thank the reviewer for their kind comments. This is an important point also raised by reviewers 1 and 2, which we have addressed by significantly expanding our findings that compare our structures with those previously published:

1. We have performed a more in-depth comparison of the structural differences between our on-target structure and the previous pre-catalytic Cas9 state (PDB 6O0Z). We had previously noted key differences in the position of the HNH domain (15 Å closer to the scissile phosphate), the REC3 domain, which appears to clamp more tightly onto the protospacer, and the REC2 domain, which moves away from the protospacer to make space for the HNH domain. Following the reviewers' suggestions, we have now also noticed that the PAM-distal region of the R-loop exhibits an extended helical pitch compared to a

linear substrate (Extended Data Figure 5e). This is a key difference that is likely due to the increased flexibility of the (-)SC DNA substrate compared to the linear one. We have discussed further on Page 5, lines 26-32.

2. We also include a comparison of the OT1 structure (9H4K) to the pre-catalytic structure (7S4U, Bravo et al. (2022), PMID: 29123204). The Cas9 architecture remains overall unchanged, but the REC2 domain is visible in their structure. The key differences between the two structures are located in the PAM distal region of the R-loop. In 7S4U, the PAM-distal DNA is distorted upwards towards the REC3 domain with a local RMSD of ~ 14 Å across the TS backbone (Extended Data Figure 6e), whereas in our OT1 structure, the R-loop is already completely formed. This is consistent with the proposed idea that the R-loop forms faster on (-)SC substrates compared to linear DNA. We have included this on Page 6, lines 36-43.
3. We also include a comparison between our OT1 structure (9H4K) and structures containing seed-region purine clashes (A:A, G:A and G:G in Pacesa et al (2022) Cell, PMID: 36306733). While the overall Cas9 architecture remains nearly identical, in our structure, the REC3 domain appears to be more outward by ~ 5.4 Å than in any of the four seed-region purine clash structures (PDBs 7QR7, 7QQR, 7QQQ, 7QQX, Extended Data Figure 6i-l). Although the REC3 domain has been implicated in R-loop proofreading (Skeens et al. (2024) Sci Adv., PMID: 38446895), Cas9 cleaves all these linear off-targets at least 10-fold slower than OT1 under (-)SC conditions. Therefore, these results are consistent with the idea that mismatches on (-)SC substrates are more tolerated than on linear DNA. We have included this on Page 6, lines 36-43.
4. We also compare the OT2 structure (9H4L) with three structures containing 3 to 5 PAM-distal mismatches in the same study (Pacesa et al., PDBs 7QR7, 7QQW, 7QR0). Again, while the overall Cas9 structure remains similar, the REC3 domain in our structure appears to be ~ 5.4 Å more outward than in any of the three PAM-distal mismatches structures (Extended Data Figure 7f-h). However, Cas9 cleaves these linear off-targets at least 10-fold slower than OT2 under (-)SC conditions, consistent with the idea that PAM-distal mismatches on (-)SC substrates are more tolerated than on linear DNA. We have included this on Page 11, lines 4-6.
5. Finally, we added a comparison of our OT1 product structure (9H4M) with the existing product structure with a linear substrate (7S4X, Bravo et al. (2022) Nature, PMID: 35236982). The overall Cas9 architecture remains the same between both structures, consistent with the fact that Cas9 holds onto the DNA substrate post-cleavage (Extended Data Figure 8g and Page 7, lines 3-4).

Together, this in-depth analysis of the various structures reveals numerous specific differences between the previous (linear and PAMer) structures and the supercoiled structures from this study, highlighting how superhelicity contributes to Cas9 function and pointing to a specific topology sensing mechanism regulated by R-loop stability and the allosteric activation of the REC and Nuclease domains.

The authors state that the “structural plasticity in the PAM distal region of the protospacer are topology dependent”. Previous work using linear dsDNA templates (Jinek 2012) has also shown that when there are mismatches between 15-20 nt downstream of the PAM and the guide, there is efficient cleavage of the substrate. So, clarification is needed on how topology further augments the ability of Cas9 to cleave DNA with complete absence of complementarity beyond 15th nucleotide from the PAM.

We thank the reviewer for raising this point. The experiments described in Jinek et al 2012 (Figure 3E therein) were performed on plasmids purified from *E. coli* (thus, negatively supercoiled), which further support our findings. We have now included this in the revised manuscript (Page 8, lines 10-13).

The experiments are done using a 126-bp long minicircle DNA. In this case, binding of one Cas9 in the circle is able to relax the circle through the process of strand exchange during the R-loop formation. Binding to an off-target site with PAM-distal mismatches prevents the relaxation. This leads to several interesting questions: (i) will the effect of Cas9 on superhelicity be same when an active Cas9 that can cleave the DNA binds to a (-)SC minicircle (several off-target sites are cut by Cas9)?;

We agree with the reviewer that this work leads to additional interesting questions. This one was already addressed in the original submission. The product structure obtained with active Cas9 shows that the relaxation is maintained even after cleavage. This is because of the exceptional stability of the product complex, as we demonstrated in our previous paper (Newton *et al.* (2018) NSMB, PMID: 30804513).

(ii) will there be an effect of the length of the minicircle, which can create different levels of topological stress, on Cas9's function?

This is also an interesting question. To address this, we have prepared larger minicircles (211 bp) with lower supercoiling density (-0.099). The cleavage data with the larger minicircles confirm that the off-targets are only cut when the DNA is negatively supercoiled (Figure 4e and Extended Data Figure 7j, and Page 8).

(iii) Will the non-requirement of the positions beyond base pairs 12-14 change based on length of the minicircle and the ability of Cas9 to cleave DNA? Cas9 can sometimes nick off-target site, though not linearise, and this may also contribute differently to the superhelicity of the DNA. Articulation of these facts will elevate the mechanistic aspects derived from this work.

To address this, we used the larger minicircles (211 bp) with truncated guides. The resulting cleavage data confirm the same non-requirement beyond base pair 14 (Figure 4e and Extended Data Figure 7j and Page 8).

The conclusion that (-)SC causes HNH to swing closer to the active state may need further validation. The present analysis is based on PDB: 6O0Z. The authors then mention that in PDB 4OO8, a similar positioning can be seen as was observed in their structure and that partial stranded DNA substrate used in PDB 4OO8 may be similar to (-)SC DNA. This raises the question if the HNH placement is due to

topological changes of the substrate or due to the fact that the DNA is single-stranded near the cleavage position. Both references 15 and 16 have a series of structures reported. Similarly, there are several other Cas9 structures that represent different conformational states of the complex. A systematic analysis of a set of selected structures that has used different types of DNA substrates (e.g., dsDNA, partial dsDNA, different lengths of DNA, presence/absence of divalent metal, active vs. dead Cas9) may provide details on how the positioning of HNH depends on different features of the DNA substrate.

This is an important point that we have addressed by expanding our discussion to compare our pre-catalytic structure (9H4J) with several previously published ones with distinct DNA features (Response Figure 2: 6O0Z, with a dsDNA substrate; 5F9R, with a partial dsDNA substrate; 4UN3, with a PAMer substrate; 4O08, with ssDNA only).

In the initial submission, we already compared our on-target structure with 6O0Z and highlighted key conformational differences in the HNH domain (15 Å closer to the scissile phosphate in our structure). We also note now that both the REC2 and REC3 domains appear to be ~5 Å more outwards than in 6O0Z, consistent with the idea of a more relaxed R-loop checkpoint in the supercoiled structure. In the other three structures (5F9R, 4UN3 and 4O08), the HNH domain appears in a similar position to our structure. However, the REC2 and REC3 domains also appear to be more outwards than in our structure, consistent with the idea of a more relaxed R-loop checkpoint in the supercoiled structure. We have now included this in the revised discussion (Page 10, lines 14-19 and Extended Data Figure 5f-i).

Response Figure 2. Structural comparison of Cas9 structures across different DNA substrates.

a. On-target (9H4J) structure (this study) versus 6O0Z, both substrates consist of dsDNA. b. On-target (9H4J) structure) versus 5F9R consisting of a partial dsDNA substrate. c. On-target (9H4J) structure versus 4UN3 which consists of a PAMer substrate. d. On-target (9H4J) structure versus 4O08 which contains a single-stranded DNA substrate.

Some discussion on how the superhelicity affects Cas9's function would be beneficial. For example, is the ssDNA region in the (-)SC DNA helping in loading of Cas9 and target search or helping in strand separation for R-loop, or both?

The FRET experiment may provide more insights here based on the initial loading position vs. the PAM site with respect to different binding events that were observed.

This is a very interesting point that we are currently investigating using SHAN-seq (Morgan et al. 2024 NAR, PMID: 39106172) in collaboration with Keir Neuman (NIH). The data are still too preliminary to draw substantial conclusions, but the ssDNA regions in the (-)SC DNA appear to be located away from the PAM sequence and the protospacer, suggesting that the loading is not driven by local defects in the DNA structure (e.g., kinks). We believe that addressing this point in detail is beyond the scope of the current manuscript. But it will be addressed when the SHAN-seq studies are completed.

The minicircle DNA sequence seem to have a high A-T rich content. How may this affect the superhelicity of the minicircle and was the design strategic to create a minicircle that can be negatively supercoiled by gyrase?

The reviewer is correct. AT tracts were positioned to promote bendability. This is a commonly used strategy to prepare small DNA substrates (Crothers et al. (1990) J Biol Chem. PMID: 2185240, Pasi et al. (2016) NAR. PMID: 27439712, Cui et al. (2018) RSC Adv. PMID: 35539641, Gasiunas et al. (2020) Nat Commun. PMID: 33139742). We have now clarified this in the revised Extended Data (Methods section, Page 2, lines 32-33).

Several types of mismatch accommodation with modified base pairing are reported in the structures bound to OT1. Of these, the G:G mismatch at position 3 has an opposite orientation of the base compared to what was observed in ref 16. Is this a random chance or it has any specific relation to (-)SC?

This is an interesting question that would require a large number of G:G mismatch structures to be answered thoroughly, which is beyond the scope of this manuscript. There are also no other G:G mismatch structures in the literature to further compare with our structure, and the structure in Ref. 16 does not contain a NTS (PAMer substrate).

However, we can provide a partial answer here. Careful inspection of the OT1 product structure (9H4M, Extended Data Figure 8i) reveals that the G:G mismatch orientation has completely rearranged following backbone cleavage between positions 3 and 4. In the pre-cleavage configuration, the rG base is in an anti-configuration relative to the sugar, whereas the dG base is in a syn-configuration (Figure 3d). Once the backbone is cleaved, both the rG and dG bases appear in syn-configurations. This post-cleavage rearrangement is consistent with the idea that the pre-cleavage configuration is (at least in part) stabilised by the protospacer structure in the context of the (-)SC substrate, and therefore not a random positioning of the base. To clarify this point, we have expanded our discussion (Page 7, lines 14-17).

Ref 16 shows that mismatch at position 13 has dG in anti conformation, while a mismatch at position 11 has dA in syn conformation. This shows that either the RNA or the DNA base can take an anti or syn conformation in different mismatches. A deeper analysis of the different mismatches, its position in relation to PAM in both

refs 15 and 16 may provide the diversity of such non-canonical base pairing and if this is a general phenomenon or if some of the base pairing is specifically facilitated by the (-)SC of the substrate.

The reviewer is correct. However, we believe that comparing G:G mismatches with A:G mismatches can be unwarranted because their base-pairing faces exhibit very different chemical properties: G has two H-bond donors (NH₂ and NH) and one acceptor (exocyclic O), whereas A has one donor (NH₂) and one acceptor (cyclic N). This can result in very different mismatch pairing geometries. Furthermore, the structures in Ref 16 lack an NTS (PAMer substrates), which affects the DNA structure outside the Cas9 complex by strand reannealing. However, to address the reviewer's comment, we have clarified this in the revised manuscript (Page 10, line 49 to Page 11, line 3).

An analysis of the flexibility of REC2 and HNH in these structures will also provide more details on how the allosteric regulation of REC2 and HNH domains are different when bound to (-)SC off-target DNA compared to a linear DNA.

It is challenging to make meaningful direct comparisons of the flexibility of REC2 and HNH in these structures with ours because (i) the structures in Ref 16 lack the NTS (PAMer substrates), which affects the interactions of the NTS with the Nuclease domains and DNA structure outside the Cas9 complex; (ii) the number of total mismatches in Ref 16 is much lower than ours (3 vs 5 in OT1); and (iii) the mismatches appear in different nearest-neighbour contexts.

However, in an attempt to address this comment, we have analysed the B-factors for these structures. In either case (A:G mismatch at position 11 and G:G mismatch at position 13), the b-factors of both the REC2 (117 and 89 Å², respectively) and HNH (93 and 76 Å², respectively) domains are higher than the average b-factors of the structure (79 and 68 Å², respectively), consistent with the idea that these domains are more dynamic than the rest of the protein, and with our single molecule data (Figure 5). To address the reviewer's concern, we have clarified this in the revised manuscript (Page 6, lines 14-17).

OT1 has 5 mismatches covering positions 3, 8, 9, 15, 20 (including seed and PAM distal), while OT2 has only 4 mismatches covering positions 8, 15, 16, 20 (lesser number of seed mismatches). The PAM-distal cryo-EM density in OT2 is also poor indicating flexibility in this region. The authors state that having three mismatches at the PAM-distal end maybe causing this flexibility. It can also be interpreted that having three PAM-distal mismatches with the RNA in OT2 may make the dsDNA more stable at this end, compared to having only two mismatches in OT1.

The reviewer raises an interesting possibility. But we don't think the data supports this option because the DNA density is also lacking (compared to the on-target and OT1). If the DNA was more stable at the PAM distal end, it would be visible.

It is also stated that "The absence of discernable density in the nine PAM-distal bp of the R-loop raises the interesting possibility that the increased flexibility of the (-)SC DNA bases decreases the requirement of a complete R-loop formation for cleavage activity." Related to this, ref 16 has used only a short NTS strand and a full-length

TS strand for crystallization, and this partial dsDNA may give a more flexible PAM-distal end compared to having a (–)SC. How different are the structures of OT2 compared to some of the off-targets from ref 16?

Ref 16 also states that there is unpairing at the PAM-distal end DNA when there are mismatches at this region. In short, a thorough analysis of the diverse Cas9 structures with mismatches will help in delineating the differences that are arising exclusively from the superhelicity of the DNA.

We have already addressed these two points earlier (see Page R9). For simplicity, we duplicate the response here:

We also compare the OT2 structure (9H4L) with three structures containing 3 to 5 PAM-distal mismatches in the same study (Pacesa et al., PDBs 7QR7, 7QQW, 7QR0). Again, while the overall Cas9 structure remains similar, the REC3 domain in our structure appears to be ~5.4 Å more outward than in any of the three PAM-distal mismatches structures (Extended Data Figure 7f-h). However, Cas9 cleaves these linear off-targets at least 10-fold slower than OT2 under (–)SC conditions, consistent with the idea that PAM-distal mismatches on (–)SC substrates are more tolerated than on linear DNA. We have included this on Page 11, lines 4-6.

The AFM data on binding of dCas9 to different minicircles may need further explanation. In the main figures, only one type of minicircle is shown, which maybe that of the on-target DNA (the legend is not clear on this). In ED Fig. 3, there is data for on-target, OT1 and OT2, but the legend or the main text does not explain the conclusion from the data.

We thank the reviewer for bringing this to our attention. We have now further clarified these data in the corresponding Figure legends (see Fig. 1 and Extended Data Fig. 2 and 3).

Is there any difference in the amount of DNA relaxation between OT1 and OT2 sites?

To address this point, we have performed additional curvature analysis from the AFM images. This analysis shows a significant difference ($p=0.033$, T-Test) in the curvature at the apex of the minicircle for OT1 compared with OT2. We define this as the central mean curvature, which is measured along 10 nm of the minicircle furthest from the bound dCas9 protein, (see Extended Data Fig. 3r-t). Though there are similarities in the global distributions for OT1 and OT2, we see a large number of molecules with higher mean curvature for OT2, consistent with a structure that has been only partially relaxed. We have now clarified this point on Page 7, lines 31-33.

There are not many binding events seen in panel c for OT2.

We thank the reviewer for spotting this. We have now picked an image with more binding events, and included the number of particles analysed in the caption, which was missing in our initial submission (see Extended Data Fig. 3).

In ED Fig. 8f, the amount of nicking for sgRNA with only 14-nt guide is very low. The nicking is proposed to be related to the 14-nt bubble causing off-target as measured by CIRCLE-seq in ref 24. Nicked DNA can be repaired by human cells, and hence only nicking potential may not directly relate to off-target effects. The statement “In recent linear off-target structures 15,16, PAM-distal mismatches were clearly resolved, supporting the idea that Cas9 cleavage activity with short guides (~14 nt) is specific to (-)SC DNA substrates.”, is not clear. How is resolution of the base pairs related to DNA cleavage efficiency?

Our wording may have caused some confusion here. We have removed this statement for clarity.

How are the docking positions of the HNH different than what was previously reported using truncated RNAs (DOI: 10.1126/sciadv.aao002)? Some direct comparisons of the previous results are needed to show the effects coming only from (-)SC. A 17-nt long checkpoint was determined in this previous work, which showed that only after reaching this position, HNH can achieve the activated state. Will the (-) SC reduce the number of base pairs needed for this checkpoint?

To address this, we have now performed optical tweezers experiments at superhelical density $\sigma \sim -0.1$ using truncated gRNAs (from 18-14 nt) targeting $\lambda 2$. The resulting kymographs and smFRET analysis (Extended Data Figure 10g-i) show that many of the off-target bound molecules can readily reach a high FRET state (≥ 0.8), at least transiently, as previously reported on linear substrates (DOI: 10.1126/sciadv.aao0027), indicating that the HNH domain can reach similar docking positions.

As suggested by the reviewer, these results are consistent with a reduction of base pairs needed for this checkpoint under supercoiled conditions from position 17 in linear substrates, to position 14 on (-)SC substrates based on the cleavage data (Figure 4e, Extended Data Figure 7i-j) and smFRET data (Extended Data Figure 10g-i). This is an important point that we have now clarified in the text (Page 11, lines 36).

Does this number differ based on on-target and off-target DNA substrates in the case of (-) SC and also based on the length of the minicircle?

To address this, we performed cleavage assays using truncated guide RNAs on the larger 211 bp (-)SC DNA minicircles. The resulting data (Figure 4e, Extended Data Figure 7i-j) show the same cleavage pattern as with the smaller 126 bp (i.e., a 16 nt truncated gRNA is sufficient to cleave). We have now addressed this in the text (Page 8).

The statement, “These results show that Cas9 high-fidelity variants derived from structures in complex with linear DNA substrates are unlikely to maintaining their high fidelity in the context of (-)SC DNA, thereby justifying the use of (-)SC minicircles to further improve Cas9 fidelity.”, may need some modification. Several of the high-fidelity variants were developed by direct evolution which does not specifically modify positions based on structural interaction. In addition, these variants are tested for their ability to edit genes in cell lines, which represent native

genomic organization with (-) SC DNA. So, the directly evolved high-fidelity variants are being selected based on real interactions with cellular DNA, not based on information from a molecular structure.

We agree with the reviewer's concerns. The original claim was just too broad. To address this, we have removed this sentence for clarity.

Showing an overlay of the new structures with some from ref 16 may help to assess if the movement of the domains are due to only topology or the different structures are being captured at different intermediate steps towards reaching the precatalytic state.

We agree with the reviewer's suggestion. We have now added a new figure overlaying our structures with several structures from Ref. 16 (Extended Data Figure 6 and 7).

Minor points

Fig. 3: Show positions for HNH and REC3, maybe using dashed circles?

Done.

Please specify if Fig. 4a is OT1 with or without Mg²⁺.

All structures in this study are in the presence of 10 mM MgCl₂. We have now clarified this in the figure legends.

In Fig. 3, HNH is not visible, in 4a it is. It is better to clarify this in the legend since there are two OT1 structures being reported.

We thank the reviewer for pointing this out. The legends of Figures 3 and 4 were not clear. To address this, we have rewritten those legends for clarity.

Is OT2 pre- or post-cleavage? If OT2 is pre-cleavage, why are the post- and pre-cleavage stages being compared in this figure? It should both be the same state to show the difference in domain placements due to the effect of off-target at different positions of the protospacer.

There seems to be some confusion here. OT2 is a pre-cleavage structure and is only compared to ON and pre-cleavage OT1 in the text. We have clarified this the revised manuscript to avoid confusion (Page 7, line 28-29).

Extended data shows different maps, labelled map #1, 2, 3, 4, 5 etc. for different figures. Map 3 is listed in both ED Figs 6 and 7. Are these maps from different population of particles? The numbering scheme is not clear.

We thank the reviewer for spotting this. We have now updated the Extended Data Figures to list which figures relate to each map (e.g., Map #1 relates to Figure 2b).

In ED Fig. 11, what is the time for panel e?

We assume the reviewer was referring to the cleavage assay in Extended Data Figure 12e, where we had neglected to add the cleavage time (20 min). We thank the reviewer for spotting this. It has been clarified in the figure legend accordingly.

For ED Fig. 12, please explain the image and corresponding graph that shows different binding events, specifying how they relate to different variants and off-target binding. This will help a non-expert to understand this data better.

We have now clarified this in the figure legend.

Did the DNA used in this experiment hold both on-target and off-target sites?

We used a guide targeting the $\lambda 2$ site on λ -DNA as described in Sternberg et al (2014) Nature (PMID: 24476820). It contains one on-target site and multiple possible off-target sites as described in Newton et al (2019) NSMB (PMID: 30804513) and (2023) Mol Cell (PMID: 37802026). We have now clarified this in the methods section.

How does the on-target binding compare to off-target binding in the different variants?

We have re-analysed the high-fidelity Cas9 binding times (Extended Data Figure 10), and found that both high fidelity variants (Sniper2L and SuperFi) exhibit similar average bound times (4 ± 1 s) to the WT Cas9 (5 ± 1 s). We have now clarified this in the text (Page 9, lines 19-20)

Referee #4:

Smith et al. present in their manuscript a beautiful set of structures of Cas9 bound to initially negatively supercoiled minicircles where even the minicircles are resolved. Even from an esthetic point of view it is fun to see this. As main findings they show that for fully matching target DNA the HNH domain engaged closer to the TS scissile phosphate compared to relaxed DNA. Furthermore they present two structures on off-targets containing multiple mismatches, one with a full R-loop (OT1) and one with a partial R-loop (OT2). For some of the resolved mismatches they obtain new configurations that have not been observed before. Furthermore, they use fluorescence experiments on supercoiled DNA to show that the HNH domain is highly dynamic on supercoil-induced off-targets. Overall this is a very interesting and carefully executed study which is also very well presented. I have therefore only very few technical remarks. I am convinced that the work will be highly interesting for the Cas9 community, in particular to researchers focusing on the structural aspects of Cas9.

Detailed remarks:

- How relevant are the applied supercoiling densities in this study (absolute values of ≥ 0.1) compared to supercoiling levels in eukaryotic systems where the DNA is rather relaxed (in contrast to prokaryotic systems with supercoiling densities of around -0.05)? Are there any reliable quantitative numbers for the supercoiling densities in eukaryotes in order to evaluate whether supercoiling can provide a detectable impact on off-targeting in these systems?

The distribution of positive and negative supercoiling in mammalian cells has been mapped by relative genome-wide psoralen binding (Kouzine et al., 2013; Naughton et al., 2013). We are not currently aware of any studies directly reporting the absolute supercoiling density of DNA in mammalian cells. However, this can be estimated. It is known that *in vitro* psoralen binding is 1.7x higher to DNA with a supercoiling density of -0.065 , and that this relationship is roughly linear (Kramer et al., 1999). The genome-wide psoralen mapping studies report psoralen binding levels ~ 1.4 times higher than in relaxed DNA at many loci (Naughton et al., 2013), corresponding to a supercoiling density of ~ -0.055 at some sites. While this is a very rough estimate, it shows that we are working close to physiological levels, especially with the new cleavage data with larger minicircles at -0.1 supercoiling densities (Figure 4e, Extended Data 7j). In addition, experiments described in Jinek et al 2012 (Figure 3E therein) with plasmids purified from *E. coli* (thus, ~ -0.07 supercoiling densities) are consistent with our results, further supporting our findings. We have now clarified this in the revised Discussion (Page 11, lines 43-48 and Page 12, lines 1-4).

- Are the unique mismatch configurations on OT1 and OT2 observed in this study really supercoil-specific or would they also be observable on relaxed DNA when using the same base-pair neighbors? In other words, is the occurrence of the new mismatch configurations just caused by the so far limited number of available mismatch structures?

This is an important point that was raised by other reviewers. To address this, we have performed cleavage assays on linearised substrates (see Extended Figure 1f-i). The data demonstrate that Cas9 does not cleave either OT1 or OT2 in linear form, supporting our main hypothesis that (-)SC induced off-target cleavage is not caused by the specific sequence of OT1 or OT2.

Moreover, we have performed additional experiments with two new off-targets (OT3 and OT4) with distinct mismatch distributions (Extended Figure 1). These data further support the conclusion that (-)SC induced off-target cleavage is a general effect. We have now clarified this in the revised manuscript (Page 3, lines 31-35 and Extended Figure 1f-i).

- The authors state that "negative supercoiling reduces the potential energy barriers to unwind the DNA substrate (dsDNA) and facilitate R-loop formation, even in the presence of multiple mismatches". In this context they should refer to previous studies on CRISPR-Cas effectors that evaluated quantitatively how mismatches are overcome as a function of the applied supercoiling and how this can be effectively modelled/understood.

We agree with the reviewer. To address this, we cite the work of Ivanov et al. (2019) PNAS (PMID: 32123105) and Szczelkun et al. (2014) PNAS (PMID: 24912165).

- The HNH dynamics is still somewhat confusing to me. While I understand that the HNH domain can be well resolved in structures with a fully matching target, I don't understand why it appears to be flexible for the uncleaved OT1 target despite the presence of a full R-loop. Are there any specific contacts of the HNH domain to the heteroduplex that would be not established if the heteroduplex contains mismatches?

We agree with the reviewer that it can seem confusing that, given that both R-loops are formed and both minicircles are cleaved by Cas9, the HNH domain in the on-target structure appears better resolved than in OT1.

Comparison of the two structures shows that the REC2 domain is also more mobile in the OT1 structure. This points to a previously described allosteric activation mechanism regulated by R-loop stability and HNH/REC dynamics, and is consistent with the lower residence time of the HNH domain in the high FRET state of off-targets (Figure 5). In turn, this can readily explain the slower cleavage kinetics of OT1 (Figure 1c).

To further clarify this in the manuscript, we have expanded the Discussion accordingly (Page 11, lines 7-13).

- Ext. Data Fig. 11: D) The obtained decay time of $\tau = 0.6$ s seems to be inconsistent with plotted data and fit function. At τ the number of FRET molecules should have decreased to $1/e$ (about $(1/3)$ of the initial amplitude. Neither data nor fit show this, however. The errors for τ are in my opinion too high in D) but too low in F) given the high and low event numbers, respectively. The standard deviation of a normalized exponential distribution is τ , such that the standard error of τ should

be given by τ/\sqrt{N} . This way the error in D should be about 0.06 s while in F about 1s. The authors should then test whether the difference of tau for the two different event types is still significant.

The confidence intervals reported in these Figures represent the standard errors of the fit, as determined by Igor Pro 8 (Wavemetrics). The calculated p-value for these values is $p < 0.0001$ (t-test, following Ref 1). To address this, we have now included the p-value in the caption of Extended Data Figure 9.

1. Kirkwood BR, Sterne JAC (2003) Essential medical statistics, 2nd ed. Oxford: Blackwell Science

Referee #1:

The authors have addressed all of our questions, and I have no further major concerns.

We thank the reviewer for their comments and contributions towards this manuscript.

1. On p. 3, line 25, "...because (-)SC the DNA substrate..." should be revised to "...because (-)SC DNA substrate...".

Done.

2. In Response 2, the reference to "the text on Page 11, lines 25-27" appears to be incorrect.

Corrected to page 12, lines 28-34.

3. In Response 5, the statement "The data show that relaxed DNA minicircles do not form stable R-loops (Extended Data Figure 1j)" should be updated to refer to "Extended Data Figure 1l,m."

Done.

Referee #3:

The authors have made a tremendous effort in addressing my queries from the first submission. Comparison of the new structures with several previous ones truly clarifies the effects on superhelicity on on-target and off-target DNA cleavages. I also appreciate the added biochemical and binding studies that also confirmed the effect of superhelicity on DNA binding events as well as on the relaxed checkpoint needed for negatively supercoiled DNA compared to linear DNA. I do not have any major comments, just one below to check if a figure citation is correct.

We thank the reviewer for their comments and contributions towards this manuscript.

Minor comments:

Organizing figures in the order of appearance in the main text will improve readability. Ext. Data Fig. 8 is cited in the main text before Ext. Data Fig. 7.

We've changed the order of Ext. Data Figure 7 & 8 to improve readability.

Please check lines 29-31 of p. 6 ("While the entire DNA ring is still visible, it now adopts a 'teardrop' conformation (Extended Data Fig. 8b), indicating that the DNA is not fully relaxed upon dCas9 binding, consistent with partial R-loop formation."). Is this Ext. Data Fig. 8b or Ext. Data Fig. 7b? Switching current Ext. Data Fig. 7b to Ext. Data Fig. 8b will make the flow align better with the order things are mentioned in the main text.

We've changed the order of Ext. Data Figure 7 & 8 to improve readability.

Referee #4:

The authors have well addressed my previous remarks/concerns such that I recommend publication of the manuscript.

We thank the reviewer for their comments and contributions towards this manuscript.

The only point that was not correctly addressed is my previous final remark: Ext. Data Fig. 11: D) The obtained decay time of $\tau = 0.6$ s seems to be inconsistent with plotted data and fit function. At τ the number of FRET molecules should have decreased to $1/e$ (about $(1/3)$ of the initial amplitude. Neither data nor fit show this, however. The errors for τ are in my opinion too high in D) but too low in F) given the high and low event numbers, respectively. The standard deviation of a normalized exponential distribution is τ , such that the standard error of τ should be given by τ/\sqrt{N} . This way the error in D should be about 0.06 s while in F about 1s. The authors should then test whether the difference of τ for the two different event types is still significant.

While the authors have corrected the values for τ in both plots, the error in the Ext. Data Fig. 9h does still not make sense. It is impossible to obtain an error of 0.02 s for an exponential decay with a mean of 4.87 s from just 19 measured values. Least mean square fitting tends to underestimate the errors since it uses as data errors the actual rms. I explained previously how the error could be estimated. In the light of this I doubt that the obtained p value of 0.0001 is correct. When using a t-test, the authors should consider that the two distributions have very different sample sizes and also different standard deviations. Furthermore, the p value is not provided in the caption of Ext. Data Fig. 9 as promised by the authors.

We thank the reviewer for spotting this error on our part. We reported the wrong errors here. It should have been 1.6 ± 0.2 s and 4.9 ± 2.0 s, respectively. We have now fixed this in the final figures. Regarding the p-value, making sure we use the correct values, we have calculated the p-value as follows:

First, we calculate the pooled standard deviation s :

$$s = \sqrt{\frac{(n_1 - 1)s_1^2 + (n_2 - 1)s_2^2}{n_1 + n_2 - 2}}$$

Where s_1 and s_2 are the standard deviations of the two taus (0.2 and 2.0 s, respectively) with sample sizes n_1 (= 125) and n_2 (= 19).

The standard error se of the difference between the two values is calculated as:

$$se(\bar{x}_1 - \bar{x}_2) = s \times \sqrt{\frac{1}{n_1} + \frac{1}{n_2}}$$

The p-value, is calculated using the t-test, with the value t calculated as:

$$t = \frac{\bar{x}_1 - \bar{x}_2}{se(\bar{x}_1 - \bar{x}_2)}$$

The p-value is the area of the t distribution with $(n_1 + n_2 - 2)$ degrees of freedom, that falls outside $\pm t$.

We obtain a p-value < 0.0001 . A p-value < 0.05 , indicates that the two taus are significantly different. We have now included this p-value in the caption.